# Some Fundamental Aspects about Lipschitz Continuity of Neural Networks

Grigory Khromov[*][a] and  Sidak Pal Singh[*][a,b]

[a] Department of Computer Science, ETH Zürich
[b] Max Planck ETH Center for Learning Systems

## Abstract

Lipschitz continuity is a crucial functional property of any predictive model, that naturally governs its robustness, generalisation, as well as adversarial vulnerability. Contrary to other works that focus on obtaining tighter bounds and developing different practical strategies to enforce certain Lipschitz properties, we aim to thoroughly examine and characterise the Lipschitz behaviour of Neural Networks. Thus, we carry out an empirical investigation in a range of different settings (namely, architectures, datasets, label noise, and more) by exhausting the limits of the simplest and the most general lower and upper bounds. As a highlight of this investigation, we showcase a remarkable *fidelity of the lower Lipschitz bound*, identify a striking *Double Descent trend in both upper and lower bounds to the Lipschitz* and explain the intriguing *effects of label noise on function smoothness and generalisation.*

## 1 Introduction

Lipschitz continuity of a function is a key property that reflects its smoothness as the input is perturbed — since, to put it more accurately, it is the maximum absolute change in the function value per unit norm change in the input. Typically, in machine learning, we desire that the learned predictive function is robust, i.e., not overly sensitive to changes in the input. If achieving point-wise robustness in the entire input domain is difficult, then we would, at least, hope to be robust over a large and representative subset of the input space with respect to the underlying data distribution (like in the vicinity of the training samples). Otherwise, we can expect that a function which varies too drastically in response to minuscule changes in the input, especially when the input is near the decision boundary or from another representative region, would struggle to generalise well on unseen test data. Also, on the flip side, we would be wary if the value of the Lipschitz constant (associated with the corresponding Lipschitz continuity) is extremely small (for the sake of the argument, think zero) over the entire input space, for this would imply an excessively large bias (Geman et al., 1992) and essentially a useless function.

This intuitively captures why the Lipschitz constant of the model function would shed light on the nature of its fit. Thus, not all too surprisingly, the Lipschitz constant has been shown to play a crucial role in various topics such as generalisation (Bartlett et al., 2017), robustness (Weng et al., 2018), vulnerability to adversarial examples (Goodfellow et al., 2015; Huster et al., 2018), and more. As a result, this has spurred a significant body of literature, especially, in producing tighter estimates to the Lipschitz (Jordan & Dimakis, 2020; Virmaux & Scaman, 2018; Xu et al., 2020; Fazlyab et al., 2019; Gómez et al., 2020) and applying different forms of explicit Lipschitz controls (Anil et al., 2018; Gouk et al., 2021; Cissé et al., 2017; Petzka et al., 2017; Tsuzuku et al., 2018).

**Aim of our study.** In contrast, our work forms a slight departure from these (mainstream) categories: we want to understand the inherent nature of the Lipschitz constant and its phenomenology as seen, in practice, in Deep Neural Networks. Yet, most importantly, we do not want to burden ourselves with tighter Lipschitz estimates that, however, restrict our investigation to small datasets like MNIST (or at most CIFAR-10) given their increased computational costs. Nor do we want to study networks where

---

[*]Equal Contribution. Correspondence to `gkhromov@ethz.ch`, `ssidak@ethz.ch`. Our code is publicly available on GitHub.

the Lipschitz was explicitly regularised in numerous distinct ways, but we want to arrive at whether there is an inherent or implicit regularity in the Lipschitz behaviour for the most common networks — that are trained without any explicit regularisation at all (Lipschitz or otherwise). Thus, our motivation stems from striving for a better understanding of modern over-parameterised deep networks, and we would like to explore and uncover fundamental aspects of Lipschitz continuity in this regard.

**Approach.** As we have discussed, on one hand, tighter estimates of the true Lipschitz are, more often than not, rather computationally expensive, which greatly limits their use. On the other hand, when simple bounds are utilised, there is an element of uncertainty about whether the findings apply to the true Lipschitz constant or just that particular bound. In this paper, however, we propose to sidestep this issue by first identifying the *simplest possible bound whose fidelity to the Lipschitz constant can be reasonably established*. Then, as a safeguard, we also track the (usual) upper bound to the Lipschitz — and thereby effectively 'sandwiching' the true Lipschitz value in between. Having done so, we then completely turn our focus to extracting as many insights as possible via these bounds, which are demonstrated on large-scale network-dataset settings (like, ResNet50 on ImageNet). This is but a simple conceptual shift in research perspective, which however lets us showcase intriguing traits about the Lipschitz constant of neural network functions in a multitude of settings.

**Contributions.** (i) We begin with investigating the nature of the Lipschitz constant, in terms of its deviation from the value at initialisation and how it behaves during the course of training. In this setup, we also show the fidelity of the local Lipschitz-based lower bound, confirm the trends with larger models and datasets, as well as provide an intuitive toy example to explain our findings. (ii) Then, we investigate (and back) the folk wisdom about over-parameterisation resulting in smooth interpolation through the lens of the Lipschitz, when networks of increasing width are trained in a Double-Descent-like setting. We supplement this empirical finding by sketching a theoretical argument for this behaviour using the bias-variance trade-off. (iii) Finally, we complete the existing picture surrounding the behaviour of the Lipschitz constant in the presence of label noise, across a wide range of network capacities and noise strengths. We find that there is an interesting interplay of network capacity, noise, functional smoothness, and memorisation that provides a more nuanced and rounded view of over-parameterisation.

Ultimately, we hope that our work facilitates further theoretical advances by providing a ground or, at least, a scaffolding to base or refute theoretical hypotheses about the Lipschitzness of neural networks.

## 2 THEORETICAL PRELIMINARIES

This paper introduces multiple bounds on the Lipschitz constant, as well as various ways of computing it. To aid the reader, we include a list of all notations in Appendix S1. We start by recalling the definition of Lipschitz-continuous functions:

**Definition 2.1 (Lipschitz continuous function)** *Function $f : \mathbb{R}^d \mapsto \mathbb{R}^K$, defined on some domain $dom(f) \subseteq \mathbb{R}^d$, is called $C$-Lipschitz continuous, $C > 0$, w.r.t some $\alpha$-norm if, $\forall \, \mathbf{x}, \mathbf{y} \in dom(f)$ : $\|f(\mathbf{x}) - f(\mathbf{y})\|_\alpha \leq C \|\mathbf{x} - \mathbf{y}\|_\alpha$ .*

Note that we are usually interested in the smallest $C$, such that the above condition holds. This is what we call the *(true) Lipschitz constant of the function $f$*, which is, unfortunately, proven to be NP-hard to compute (Virmaux & Scaman, 2018). Therefore, we focus on the upper and lower bounds of the true Lipschitz constant. To give a lower bound, we will need an alternative definition. For simplicity, we compute the 2-norm (i.e. $\tilde{\alpha} = \alpha = 2$) throughout the rest of the paper.

**Proposition 2.1 (Alternative definition (Federer, 1996; Gómez et al., 2020))** *Let function $f$ : $\mathbb{R}^d \mapsto \mathbb{R}^K$, be defined on some domain $dom(f) \subseteq \mathbb{R}^d$. Let $f$ also be differentiable and $C$-Lipschitz continuous. Then the Lipschitz constant $C$ is given by: $C = \sup_{\mathbf{x} \in dom(f)} \|\nabla_{\mathbf{x}} f\|_{\tilde{\alpha}}$ , where $\nabla_{\mathbf{x}} f$ is the Jacobian of $f$ w.r.t. to input $\mathbf{x}$. $\tilde{\alpha}$ is the dual norm of $\alpha$ if $K = 1$, otherwise $\tilde{\alpha} = \alpha$.*

**Lower bound via local Lipschitz continuity.** Instead of considering the Lipschitz constant over the entire input domain $dom(f)$, we can restrict ourselves to the subspace of the domain where the underlying data distribution is supported, and its neighbourhood (denoted as $\mathcal{D}^+$). Under the i.i.d. (independent and identically distributed) assumption in which we typically operate, this definition of *'effective Lipschitz constant'* ($C_{\mathcal{D}^+}$) carries a direct significance. Moreover, this also provides a natural way to lower bound the effective Lipschitz constant based on a finite-sample evaluation, like

on the training set $S \subseteq \mathcal{D}^+$:

$$C \geq C_{\mathcal{D}^+} = \sup_{\mathbf{x} \in \mathcal{D}^+} \|\nabla_{\mathbf{x}} f_{\boldsymbol{\theta}}\|_2 \geq \sup_{\mathbf{x} \in S} \|\nabla_{\mathbf{x}} f_{\boldsymbol{\theta}}\|_2 =: C_{\text{lower}} \qquad (1)$$

An alternative approach would be to use a straightforward Lipschitz computation: $C_{\text{alt. lower}} := \sup_{\mathbf{x}, \mathbf{y} \in \mathcal{D}^+, \mathbf{x} \neq \mathbf{y}} \frac{\|f_{\boldsymbol{\theta}}(\mathbf{x}) - f_{\boldsymbol{\theta}}(\mathbf{y})\|_2}{\|\mathbf{x} - \mathbf{y}\|_2}$. For the rest of the paper, we will focus on computing (1), as $C_{\text{alt. lower}}$ requires $O(n^2)$ evaluations, and, as shown in Appendix S2.1, results in worse estimates.

We also track $C_{\text{avg\_norm}} = \mathbb{E}_{\mathbf{x} \in S} \|\nabla_{\mathbf{x}} f_{\boldsymbol{\theta}}\|_2$, i.e., the expected value of the Jacobian norm, to better contextualise recent work. For instance, this quantity is equivalent to Geometric Complexity, introduced by Dherin et al. (2022). Yet, this can be much less than the considered lower bound estimate $C_{\text{lower}}$. Thus, on its own, it is insufficient to establish claims on the Lipschitz constant.

**Upper bound via product of layer-wise upper bounds.** Tight upper bounds to the Lipschitz constant, however, are far from straightforward to compute, as can be seen in works mentioned in Section 6. Thus, as expected, we use the simplest upper bound, which is just the product of per-layer Lipschitz constants. Let us assume we have a network $f_{\boldsymbol{\theta}}$ with $L$ layers, 1-Lipschitz non-linearity $\sigma$, such as ReLU, and parameters $\boldsymbol{\theta} \in \mathbb{R}^p$. The overall function can be then expressed as $f_{\boldsymbol{\theta}} := f^{(L)} \circ \sigma \circ f^{(L-1)} \circ \sigma \circ \cdots \circ f^{(1)}$. We can then upper bound the Lipschitz constant:

$$C \leq \prod_{i=1}^{L} \sup_{\mathbf{x}^{(i-1)} \in dom(f^{(i)})} \|\nabla_{\mathbf{x}^{(i-1)}} f^{(i)}\| \leq \prod_{i=1}^{L} \sup \|\nabla_{\mathbf{x}^{(i-1)}} f^{(i)}\| =: C_{\text{upper}}, \qquad (2)$$

where $\mathbf{x}^{(i-1)}$ denotes the input at the layer $i$, i.e., the post-activation at layer $i - 1$. In the above equation, we just used 1-Lipschitzness of the non-linearity and then considered an unconstrained supremum (see more details in Appendix S2.2). As a simple example, take the case of a single linear layer $f^{(1)}(\mathbf{x}) = \mathbf{W}^{(1)} \mathbf{x}$ — the upper bound to the Lipschitz constant is clearly equal to the spectral norm of the weight matrix, i.e., $C_{\text{upper}} = \|\mathbf{W}^{(1)}\|_2$. Similarly, for an $L$-layer network, the upper bound comes out to be the product of the spectral norms of the weight matrices: $C_{\text{upper}} = \prod_{i=1}^{L} \|\mathbf{W}^{(i)}\|_2$.

## 3 INSIGHTS INTO THE NATURE OF THE LIPSCHITZ CONSTANT

### 3.1 DOES THE LIPSCHITZ CONSTANT DEVIATE SIGNIFICANTLY FROM INITIALISATION? WHAT IS ITS EVOLUTION REALLY LIKE?

For sufficiently wide neural networks, recent theoretical works often assume that the network function and its linearised version[1] around the initialisation (Chizat et al., 2019) behave similar enough, while being identical in the limit of infinite-width (Jacot et al., 2018; Du et al., 2019). While it is still not entirely clear how close today's massively over-parameterised networks actually approach this limit, it is widely acknowledged that such a limit does not adequately capture feature learning (Arora et al., 2019; Lee et al., 2020; Samarin et al., 2022). So, it is not apparent how much is the eventual Lipschitz constant of the network determined by that at initialisation. Moreover, if it does deviate significantly, does it in fact decrease or increase, or something completely else takes place? These are the kinds of questions we would like to settle in this section.

**Evolution of the Lipschitz constant.** Let us start simple by first exploring how the Lipschitz constant evolves when training a fully-connected network (FCN) with ReLU activations (FCN ReLU for short) using cross-entropy (CE) loss. Figure 1 shows this trend for a pair of networks with moderate and extreme hidden layer widths (i.e., 256 and 65,536 respectively). We can observe that both the $C_{\text{lower}}$ and the $C_{\text{upper}}$ keep increasing as the training proceeds. In particular, while they start from similar values at initialisation, they rapidly drift apart from each other. We explore this effect through the prism of network linearisation in Appendix S3.1.

The Lipschitz behaviour seems to saturate at some point sufficiently further into the training process, but relative to the initialisation, the lower and the upper bound grow, for both widths, by a significant factor. This strictly increasing behaviour of the Lipschitz might be attributed to the nature of the CE loss, for which convergence requires taking the parameters norms to infinity. But, as a matter of fact, we find that similar behaviour can be seen in the case of MSE loss in Figure S9. So, while the much wider network has a smaller increase in Lipschitz, the growth is significant. Thus, it seems that even at such high widths, the network deviates far enough from its initial state, and the eventual Lipschitz constant overshadows the one at initialisation.

---

[1]i.e., linearisation in the function space, $f(\mathbf{x}; \boldsymbol{\theta}) \approx f(\mathbf{x}; \boldsymbol{\theta}_0) + \langle \boldsymbol{\theta} - \boldsymbol{\theta}_0, \nabla_{\boldsymbol{\theta}} f(\mathbf{x}; \boldsymbol{\theta}_0) \rangle$

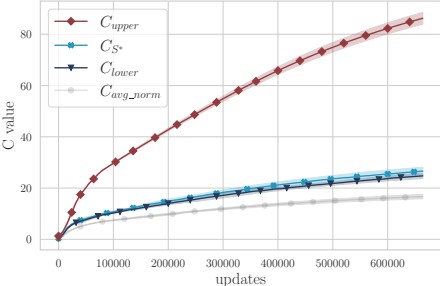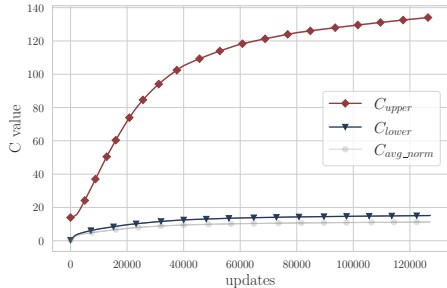

Figure 1: Plot of Lipschitz constant bounds by training epoch for **FCN ReLU network** with hidden layer widths 256 (**left**) and 65,536 (**right**) on MNIST1D. $C_{\text{upper}}$, $C_{\text{lower}}$ and $C_{\text{avg\_norm}}$ are computed on train dataset $S$, whereas $C_{S^*}$ is the local Lipschitz computed on the $S^*$. Relative to initialisation, the lower bound at convergence grows by a factor $63\times, 40\times$, while the upper bound by $66\times, 10\times$, for the widths $256; 65,536$ respectively. Results are averaged over 4 runs. See Appendix S2.6.8.

**Fidelity of the lower bound to the effective Lipschitz constant.** Next, as the effective Lipschitz constant lies somewhere between the upper and the lower bounds, what still remains unclear is how loose are these bounds and where our quantity of interest lies. To give more insight into this, we compute the lower Lipschitz bound (1) on a larger set of examples $S^*$, which is the union of the training set $S$, *the test set $S'$*, and a set of *random convex combinations* of samples from the train and test sets (see S2.6.8 for more details). The results are presented in Figure 1, corresponding to the legend $C_{S^*}$, where it becomes evident that the bound for the Lipschitz constant computed on this expanded set $S^*$ lies much closer to the lower bound $C_{\text{lower}}$ than the upper bound $C_{\text{upper}}$. While the gap between the upper and the lower bound may not seem that drastic here, the upper bound can quickly become extremely large for deeper models (c.f., Figure 2 and Appendix S3.8), leading to a significant overestimation of the Lipschitz constant. In fact, this aspect about the fidelity of the lower bound can also be spotted in some past works (Gómez et al., 2020), although it does not get a proper discussion. Overall, this seems to suggest that *the trend of effective Lipschitz constant is more faithfully captured by the lower bound $C_{lower}$, contrary to the upper bound $C_{upper}$.*

**A theoretical picture.** To understand the increasing Lipschitz trends analytically, we develop an upper bound on the Lipschitz constant in Appendix S4.1. In short, given SGD updates with a constant learning rate $\eta \in (0, \infty)$ and assuming the loss gradients are bounded by $B \in (0, \infty)$ (which could be enforced by gradient clipping, for instance), we get the that the upper bound on the Lipschitz constant $C_T$ at time step $T$ follows the trend $C_T \propto B\eta T$. Although this bound only describes the behaviour of the upper bound to the Lipschitz constant, the analysis still shows the undeniable effect of training on the function's Lipschitz continuity.

## 3.2 WHAT HAPPENS FOR BIGGER NETWORK-DATASET SETTINGS?

Given the above experiments were based on simple network-dataset settings — primarily to allow us to elucidate the point about the Lipschitz of an extremely wide Neural Network and its deviation from initialisation, we now test whether the trend about the growth of Lipschitz persists for more modern over-parametrised networks on bigger datasets as trained in practice.

To thoroughly establish whether our results carry over for networks with millions of parameters, we take the case of RESNET50 and a VISION TRANSFORMER (VIT) trained on IMAGENET. Note that we are forced to slightly restrict the computation of the Lipschitz estimates to a smaller number of checkpoints during training, as well as the number of samples. In brief, this is due to the high computational complexity [2] involved in the process of evaluating the Jacobian norms of a significant number of large matrices. The results for these experiments can be found in Figures 2, 3. As ViTs do not possess a theoretical upper bound (Kim et al., 2020) due to the presence of quadratic interactions in input and attention layers, only local Lipschitz-based estimates are presented in Figure 3.

---

[2]Computing the lower Lipschitz on the entire ImageNet would require evaluating $\sim 1.2$ million Jacobian matrices of size $1,000 \times 150,528$, for a single seed and a single checkpoint. As a reference, training a ResNet50 for 90 epochs with a 1024 batch size requires $\sim 106,000$ gradient evaluations. Hence, **lower Lipschitz estimation at a single checkpoint is almost as expensive as one entire training run**. This justifies the expediency of employing subsampling. Moreover, the variance of this procedure seems minimal (Appendix S3.6).

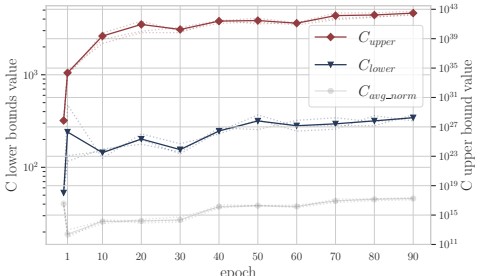 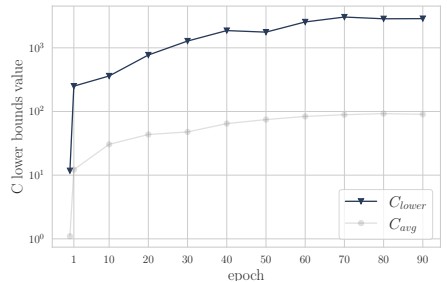

Figure 2: Lipschitz constant bounds for ResNet50 on a subset of 200,000 samples of ImageNet. Results are averaged over 3 runs. More details in Appendix S2.6.9.

Figure 3: Lower Lipschitz constant bounds evolution for ViT on a 50,000 samples ImageNet subset. More details in Appendix S3.5.

The first thing that catches our eye is that the upper bound gets almost vacuously large, with the gap between the lower and upper bounds, for instance in the case of ResNet50 in Figure 2, stretching to $\sim 40$ orders of magnitude (mind the different y-axes). This suggests that upper bounds are excessively lax (even at initialisation itself, it is of the order $1e^{27}$, which can be attributed to the exponential increase with depth). It seems more plausible that the values of the (local Lipschitz) based lower bound are more revelatory about the effective Lipschitz, or the nature of the function in general.

Given the extremely high values of the upper bound, it can be argued whether our computed lower bounds are still representative of the effective Lipschitz. To investigate this, we take the top $1,000$ samples that have the highest Jacobian norm, consider their convex combinations, and use that as a basis to evaluate the local Lipschitz. In fact, in Figure S8, we show the entire distribution of norms for these 'hard' convex combinations, as well as the entire ImageNet train set (See Appendix S2.6.10).

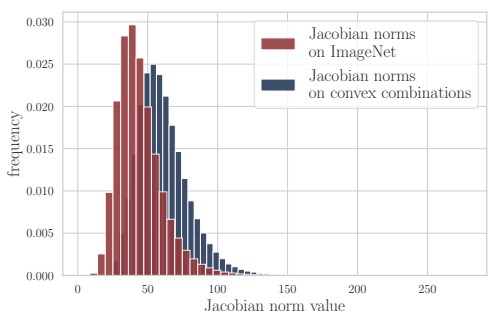

Figure 4: Distribution of the per-sample Jacobian norms for **ResNet18**, computed on the entire ImageNet and 1,000,000 hard convex combinations.

We find that while the distribution indeed shifts towards larger per-sample Jacobian norms for the hard convex combinations, the shift is not even a multiplicative factor of $2\times$ more. This shift pales in comparison to the upper bound which is over tens of orders of magnitudes higher. Overall, this strengthens our claim that the lower bound is much more faithful to the effective Lipschitz value and can hence serve better to explore various phenomena observed in over-parameterised Neural Networks. Lastly, we leave similar distribution plots for other models and datasets in Appendix S3.15.

### 3.3 GATHERING INTUITION FROM A TOY EXAMPLE

In the previous discussion, we have seen substantial evidence for the fidelity of the lower bound to the effective Lipschitz constant. In order to gather some intuition, we now consider a toy example, that gets us to the essence of our discovered findings and enhances our understanding of them.

We take a synthetic dataset on a two-dimensional closed set, for example $\mathcal{D} = [-5, 5]^2$, with 3 equally spaced classes generated by a Gaussian distribution. Due to the close proximity of the classes relative to the standard deviation of the distribution, some training points may lie in a region of another class. We now train an FCN ReLU classifier until 100% train accuracy and compute the lower and upper Lipschitz estimates, which come out to be 135.815 and 389.097, respectively. Thanks to the designed setup here, however, we can tractably compute an estimate of the effective Lipschitz constant $C_{\mathcal{D}+}$ by densely sampling 1 million points in the entire input domain and computing the maximum over the point-wise Jacobian norms. *The estimate evaluates to* 144.194, *which is rather close to the $C_{lower}$ estimate*. In fact, we can see the reason for this in Figure 5 — the *highest function change occurs near the decision boundary*, where, in turn, the local Lipschitz constant is the highest. Since the decision boundary lies in the vicinity of the training samples (this is supported by the existence of adversarial examples (Szegedy et al., 2014; Goodfellow et al., 2015)), the lower bound manages to capture the effective Lipschitz constant quite well.

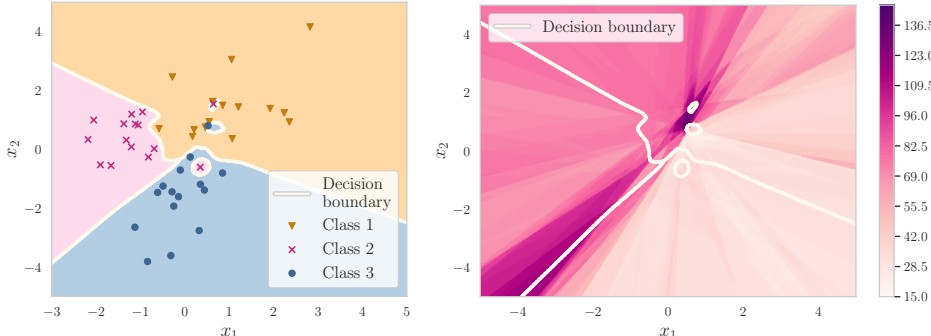

Figure 5: Plot of function prediction **(left)** and the local Lipschitz constant bounds **(right)** for the whole input domain $\mathcal{D} = [-5, 5]^2$. More details in Appendix S2.6.1.

*Adversarial and out-of-distribution (OOD) samples.* Guided by the above intuition, we also evaluate the lower bound on adversarially perturbed training samples for CIFAR-10 and OOD samples for MNIST1D. As we show in detail in Appendix S3.2 and S3.3, this evaluation only marginally improves the lower bound which does not quite justify the added computational burden — if an efficient Lipschitz estimate is the focus.

## 4   IMPLICIT LIPSCHITZ REGULARISATION, OR LIPSCHITZ DOUBLE DESCENT

A commonly held view (Belkin, 2021) about the effectiveness of heavily over-parameterised neural networks, in the context of generalisation on unseen, is that with increasing parameter count the network, in tandem with a simple optimisation algorithm like stochastic gradient descent (SGD), finds a solution where the extra capacity helps towards fitting the training samples smoothly. In contrast, with just as many parameters $p$ as the number of training samples $n$, the smoothness of the interpolation is not within control and we observe worse generalisation. And, with parameters less than that, i.e., $p < n$, the network is even unable to fully fit the training samples. Figure 6 sketches the general shape of the model in these regimes.

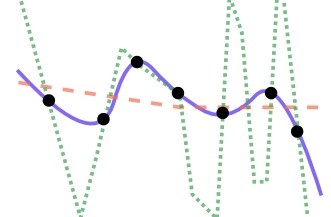

Figure 6: Cartoon of the three model regime: underfitting, overfitting, & smooth interpolation.

More concretely, this view has garnered significant evidence through the occurrence of what is known as the Double Descent (DD) phenomenon (Belkin et al., 2019; Nakkiran et al., 2020; Bartlett et al., 2020; Montanari & Zhong, 2022; Singh et al., 2022). In particular, with increasing network capacity (say, via layer width) the test loss first decreases [3], then increases up to a certain threshold of capacity (known as the interpolation threshold, where $p \approx n$), beyond which it further decreases. The theoretical works supporting double descent show — usually in fairly simplified settings — that the test loss exhibits such a behaviour. However, it largely remains difficult to attribute or pinpoint the behaviour of test loss to a core functional property of the network.

Given the above discussion about the smoothness of interpolating over-parameterised solutions, the Lipschitz constant forms a natural candidate for a functional property that signifies smoothness, and we, thus, investigate if its trend with network width has similarities to DD. In Figure 7, we conduct an experiment replicating the DD setup, by training Convolutional Neural Networks (CNNs) of increasing width (i.e., number of channels) to convergence on CIFAR-100. We notice the plot clearly shows how all three Lipschitz constant bounds grow until the interpolation threshold and decrease afterwards, while also mirroring the trend in the test loss.

Similar trends in the Lipschitz constant can also be seen for other network-dataset settings (FCNs, ViTs; MNIST1D, MNIST, CIFAR-10), as well as the MSE loss, the results for which are located in Appendix S3.9.

---

[3]This is an idealised depiction of DD, since often the initial descent requires considering very small networks and may not even be visible (Nakkiran et al., 2020; Mei & Montanari, 2022; Singh et al., 2022). The more prominent DD signature is the test loss peak at the interpolation threshold and the ensuing non-monotonicity.

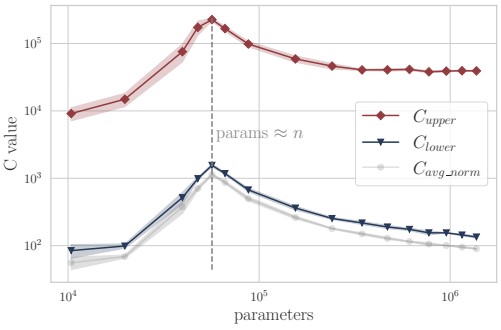 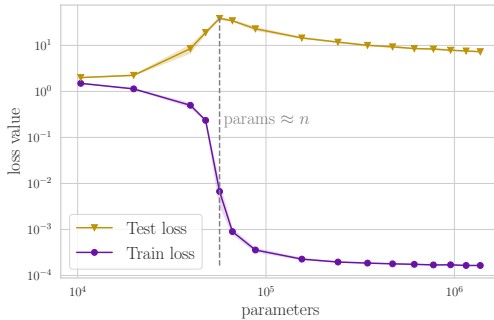

Figure 7: Comparison of various Lipschitz constant bounds with train and test losses with increasing network width, for **CNN networks** trained on **CIFAR-100** with CE loss. Results are averaged over 4 runs. More details about the networks and the training strategy are listed in Appendix S2.6.5.

**Implicit Lipschitz regularisation.** While a significant number of works in the literature specifically set out to design new and more convenient ways to explicitly regularise the Lipschitz constant during training (Gouk et al., 2021; Bungert et al., 2021) or designing architectures with Lipschitz guarantees (Anil et al., 2018; Wang et al., 2021), the above finding highlights that, even without such explicit controls, *over-parameterisation seems to provide an implicit pressure towards Lipschitz regularisation.* This could also potentially hint why, despite copious work, the direction of explicit Lipschitz regularisation has seen a relatively muted practical impact and adoption, barring certain specialised areas (Arjovsky et al., 2017; Cissé et al., 2017; Terjék, 2019). Although the strength of this implicit Lipschitz regularisation is likely problem and architecture dependent, adding explicit regularisation (here enforcing Lipschitzness), as observed in usual DD settings (Nakkiran et al., 2021), can indeed reduce the DD trend in both loss and the Lipschitz bounds (see Appendix S3.12).

Besides, the above results provide additional backing for the intuition expounded around the smoothness of interpolating solutions by Belkin (2021). Lastly, this also restores hope for the established generalisation bounds based on the Lipschitz constant (Bartlett et al., 2017) and potentially reworking them on the basis of the effective Lipschitz constant, similar to Dherin et al. (2022).

**A bias-variance trade-off argument.** Understanding the exact mechanisms of the Lipschitz DD from a more theoretical avenue would form a great direction. While this would stretch far beyond the current scope, we nevertheless supplement our discussion with a theoretical argument that connects the Lipschitz behaviour with the test loss as seen above.

The details of the analysis can be found in Appendix S4.2, but let's present the gist of the argument here. We define a Neural Network function $f_{\boldsymbol{\theta}}(\mathbf{x}, \zeta)$, where $\zeta$ indicates the noise in the function due to the choice of random initialisation (i.e. random seed). We can then bound the variance term in the test loss as follows (Var bound 1): $\mathbb{E}_{\mathbf{x} \sim S'} \operatorname{Var}_\zeta(f_{\boldsymbol{\theta}}(\mathbf{x}, \zeta)) \leq 3(\overline{C}^2 + \overline{C_\zeta}^2) \mathbb{E}_{\mathbf{x} \sim S'} \|\mathbf{x}\|^2 + \text{const}$, where $\overline{C_\zeta}$ is the mean Lipschitz constant across seeds and the Lipschitz constant $\overline{C}$ of the ensembled function $\overline{f_{\boldsymbol{\theta}}}(\cdot) = \mathbb{E}_\zeta[f_{\boldsymbol{\theta}}(\cdot, \zeta)]$. Figure 8 presents an empirical calculation for the variance bounds (results for the bias term can be found in Appendix S3.11), where the trend of the function variance aligns closely with the trend of our upper bounds, at least qualitatively.

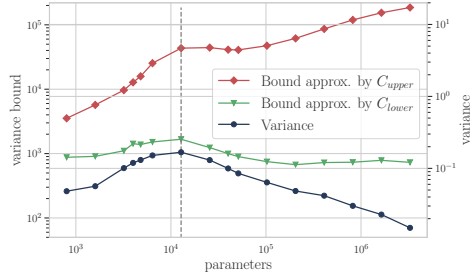

Figure 8: Variance and its upper bounds, where $\overline{C_\zeta}$ in Var bound 1 is estimated using $C_{\text{lower}}$ and $C_{\text{upper}}$ estimates. See Appendix S2.6.11.

Since the above theoretical analysis only establishes an upper bound on the function variance with the Lipschitz constant, it would be very interesting to develop a corresponding lower bound that shows the dependence on the Lipschitz constant, but that is beyond our current confines.

## 5 NOISE, CAPACITY, AND OVER-FITTING

Zhang et al. (2021) emphatically show that the over-parameterised Neural Networks have the ability to completely memorise points when trained with random labels, but these are the same networks that

also generalise when presented with clean data. However, they consider this in regard to a network with fixed capacity and a certain amount of label noise. Now imagine that we keep increasing the proportion of randomly labelled points while keeping the network fixed. Will the Lipschitz keep increasing monotonically? Will the behaviour be the same if we had a bigger or a smaller network? Introducing randomised labels should make the task harder (Bartlett et al., 2017), but does it matter as long as the network has more parameters than training samples? In this section, we explore this interplay of the noise strength, network capacity, and the resulting level of generalisation.

**Thought experiment.** Consider a task, where labels $y = 1$ if $x > 0$ and $y = -1$ otherwise, and $x$ is sampled uniformly on the reals in a certain range. Then, consider the case of an extreme amount of label noise (perhaps relative to network capacity), so that we effectively sample $y = \pm 1$ without looking at the value of $x$. In such a case, the best a network can do is to just predict $\hat{y} = 0$, $\forall \mathbf{x}$. But for such a prediction, the learned function is as smooth as it gets, with a Lipschitz constant of 0.

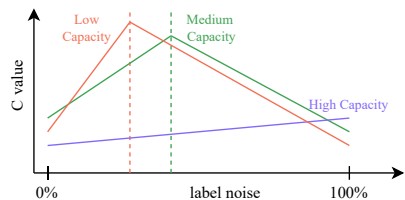

Figure 9: Outline of the behaviour of the Lipschitz constant with label noise for networks with different capacities.

**A hypothesis for Lipschitz behaviour with label noise.** Based on the intuition above, we would hypothesise that *while the network is able to fit the noise, the Lipschitz constant should increase*. At some point, when the noise strength becomes sufficiently high relative to the network capacity, we would reach a 'memorisation threshold', beyond which the predictor starts collapsing to a smoother function with a smaller Lipschitz constant. We depict our hypothesis pictorially in Figure 9.

**Empirical evidence.** To test this hypothesis, we train a bunch of CNNs on CIFAR-10 with increasing width and label shuffling levels from 0% (clean) to 100% shuffled (fully random) targets. The results can be found in the Figure 10. (i) Firstly, looking at the rows, we notice that for every noise strength, the Lipschitz constant shows a Double-Descent-like non-monotonicity with increasing width (which can perhaps be expected for low noise strengths, though not all). (ii) But, more interestingly, if we look at the columns, we find that there is a similar non-monotonicity, in line with our hypothesis above. (iii) There is a shift of the memorisation threshold, which matches the movement of the interpolation threshold in Double Descent, towards a higher parameter count in the presence of label noise (Nakkiran et al., 2020). (iv) Lastly, heavily over-parameterised networks fit random labels more smoothly, compared to networks with fewer parameters.

Overall, we see a very intriguing interplay of noise, capacity, and memorisation on the Lipschitz behaviour, which highlights, quite uniquely, the intriguing benefits imparted by over-parameterisation.

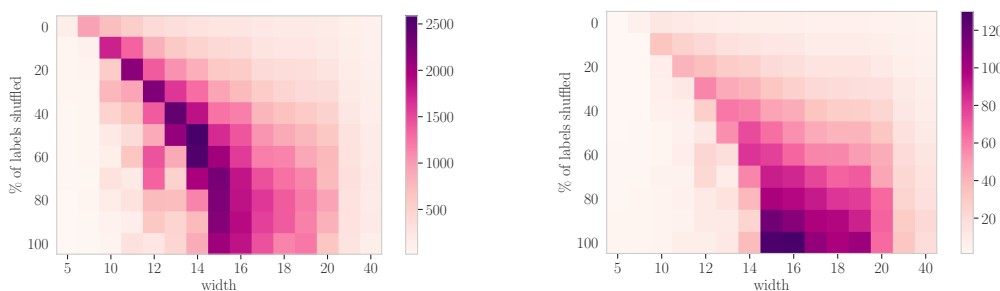

Figure 10: Lower Lipschitz values (**left**) and the test loss (**right**) for CNN models at various levels of label shuffling, trained on CIFAR-10. More details are in Appendix S2.6.16.

## 6 RELATED WORK

**Theoretical works on the Lipschitz constant.** Recent theoretical interest in the Lipschitz constant of Neural Networks has been revived since Bartlett et al. (2017) described how margin-based generalisation bounds are linearly dependent on the Lipschitz constant. Other Lipschitz constant-based generalisation bounds (Chuang et al., 2021; Munn et al., 2023) were later developed. Bubeck

& Sellke (2021) have also conjectured the phenomena of smooth interpolation in over-parametrised networks with the underlying Lipschitz continuity of the network. Since generalisation bounds and robustness guarantees (Weng et al., 2018) are upper bounded by an expression containing the true Lipschitz, extensive research has been done in the field of its accurate estimation (Virmaux & Scaman, 2018; Jordan & Dimakis, 2020; Gómez et al., 2020; Fazlyab et al., 2019; Wang et al., 2022), but this is still an ongoing pursuit as more accurate estimates typically come at high computational costs.

**Practical applications.** On the practical side, this direction can be divided into three sub-categories. *(i) Generative modelling:* The Lipschitz constant has been utilised to stabilise Recurrent Neural Networks (Erichson et al., 2021) and Generative Adversarial Networks (GANs) (Zhou et al., 2019; Petzka et al., 2017; Miyato et al., 2018; Qin et al., 2018; Zhou et al., 2018; Cui & Jiang, 2017). *(ii) Certificates for adversarial robustness:* Enforcing certain Lipschitz constraints has proved useful in certifying robustness guarantees for fully-connected Neural Networks (Anil et al., 2018; Gouk et al., 2021; Li et al., 2019) and Convolutional Networks (Cissé et al., 2017; Wang et al., 2021; Singla & Feizi, 2021; Yu et al., 2021), as well as in other areas like Equilibrium Networks (Revay et al., 2020). In order to certify certain Lipschitz values, one can simply modify internal layers so that the resulting network is $C$-Lipschitz (Kim et al., 2020), or use a more general parametrisation (Meunier et al., 2022; Prach & Lampert, 2022; Araujo et al., 2023; Wang & Manchester, 2023). *(iii) Lipschitz as a regularisation:* Another way to ensure model robustness is through Lipschitz-based regularisation techniques (Terjék, 2019; Bungert et al., 2021; Pauli et al., 2021; Tsuzuku et al., 2018; Ono et al., 2018; Leino et al., 2021). Yet, as we have seen in the DD section, there is an inherent Lipschitz regularisation already at play in Neural Networks.

**Recent works with similar focus.** Concurrent with the release of our work, Gamba et al. (2023) have also noted the connection between Lipschitz constant and Double Descent (Belkin et al., 2019), albeit by tracking only an estimate of the Lipschitz. Likewise, in the context of Double Descent and evolution during training, Munn et al. (2023); Dherin et al. (2022) explored a quantity called Geometric Complexity, which is similar to our notion of the $C_{\text{avg\_norm}}$ and the estimate of Gamba et al. (2023). But, as a matter of fact, such estimates are even looser than the local Lipschitz lower bounds, as can be seen in our results. Whereas, by considering the lower and upper Lipschitz bounds simultaneously, our results reveal the behaviour of the effective Lipschitz more faithfully. Besides, our focus is much more comprehensive as, for instance, we elaborate on the nature of effective Lipschitz (on unprecedented evaluations on ImageNet with ResNet50 and ViT) and uncover new intriguing insights into the Lipschitz behaviour in high-noise regimes.

## 7 CONCLUSION

**Summary.** In this work, we have presented a comprehensive study of some intriguing behaviours of the Lipschitz constant of Neural Networks. We first explored the nature of the effective Lipschitz constant, showcasing the evolution study (and its marked deviation from the value at initialisation) and the fidelity of the lower bound. Then, we witnessed the implicit Lipschitz regularisation effect with width in the form of Double-Descent-like non-monotonicity in the Lipschitz bounds. Finally, we examined the effect of label noise on function smoothness and generalisation, spanning across a wide range of network capacities and noise strengths.

**What we could not touch upon.** In Appendix S3, we replicate most of the experiments shown in the main part for other model classes and datasets, ensuring the consistency of presented results. Additionally, we also discuss the effect of the choice of loss (between Cross-Entropy and MSE), optimisation algorithm (SGD versus Adam), depth of the network, explicit regularisers such as weight decay and dropout, and the number of training samples.

**Limitations & future research directions.** One potential avenue for investigation would be to provide a detailed theoretical analysis for our uncovered findings which, although fell beyond the current scope, would be highly relevant (e.g. generalisation bounds based on the effective Lipschitz). It would also be rather interesting to explore the effect of large learning rates through the lens of the effective Lipschitz constant. Lastly, examining our findings outside of computer vision tasks — e.g., in the language domain, or in the framework of reinforcement learning would be a fruitful endeavor.

All in all, we hope that this work inspires further research on uncovering and understanding the characteristics of the Lipschitz constant.

## ACKNOWLEDGEMENTS

We would like to thank Thomas Hofmann, Bernhard Schölkopf, and Aurelien Lucchi for their useful comments and suggestions. We are also grateful to the members of the DALab for their support. Sidak Pal Singh would like to acknowledge the financial support from Max Planck ETH Center for Learning Systems.

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

# Appendix

# Table of Contents

# S1    TABLE OF NOTATIONS

## S1.1    BASIC NOTATIONS

Table S1: A table of basic notations.

| Notation | Definition |
|---|---|
| $C$ | Lipschitz constant |
| $f$ or $f_{\boldsymbol{\theta}}$ | Neural Network, function at hand |
| | |
| $\boldsymbol{\theta} \in \Omega$ | Parameter vector |
| $\Omega \subseteq \mathbb{R}^p$ | Parameter space |
| | |
| $\mathbf{x} \in \mathcal{D}^+$ | Input/data vector |
| $\mathcal{D}^+ \subseteq \mathbb{R}^d$ | Input/data space and its neighbourhood |
| | |
| $K$ | Number of function outputs/classes |
| | |
| $S$ | Training set |
| $S'$ | Test set |
| $\mathcal{L}$ | Loss function |

## S1.2    LIPSCHITZ NOTATIONS

Table S2: A table of Lipschitz constant notations.

| Notation of the bound | Parameter space | Input space | Formula |
|---|---|---|---|
| True Lipschitz, $C_{\text{true}}$ | at epoch $t$, i.e. $\boldsymbol{\theta}^t$ | $\mathbf{x} \in dom(f)$ | $\sup_{\mathbf{x} \in dom(f)} \|\nabla_{\mathbf{x}} f(\boldsymbol{\theta}^t, \mathbf{x})\|$ |
| Effective Lipschitz, $C_{\mathcal{D}^+}$ | at epoch $t$, i.e. $\boldsymbol{\theta}^t$ | $\mathbf{x} \in \mathcal{D}^+$ | $\sup_{\mathbf{x} \in \mathcal{D}^+} \|\nabla_{\mathbf{x}} f(\boldsymbol{\theta}^t, \mathbf{x})\|$ |
| Lower Lipschitz, $C_{\text{lower}}$ | at epoch $t$, i.e. $\boldsymbol{\theta}^t$ | $\mathbf{x} \in S$ | $\sup_{\mathbf{x} \in S} \|\nabla_{\mathbf{x}} f(\boldsymbol{\theta}^t, \mathbf{x})\|$ |
| Local Lipschitz | at epoch $t$, i.e. $\boldsymbol{\theta}^t$ | $\mathbf{x} \in dom(f)$ | $\|\nabla_{\mathbf{x}} f(\boldsymbol{\theta}^t, \mathbf{x})\|$ |
| | | | |
| Average, $C_{\text{avg}}$ | at epoch $t$, i.e. $\boldsymbol{\theta}^t$ | $\mathbf{x} \in S$ | $\frac{1}{|S|} \sum_{\mathbf{x} \in S} \|\nabla_{\mathbf{x}} f(\boldsymbol{\theta}^t, \mathbf{x})\|$ |
| Alt. lower, $C_{\text{alt.lower}}$ | at epoch $t$, i.e. $\boldsymbol{\theta}^t$ | $\mathbf{x}, \mathbf{y} \in S$ | $\sup_{\mathbf{x}, \mathbf{y} \in S, \mathbf{x} \neq \mathbf{y}} \frac{\|f(\boldsymbol{\theta}^t, \mathbf{x}) - f(\boldsymbol{\theta}^t, \mathbf{y})\|}{\|\mathbf{x} - \mathbf{y}\|}$ |
| | | | |
| Adversarial lower | at epoch $t$, i.e. $\boldsymbol{\theta}^t$ | $\mathbf{x} \in S$ | $\sup_{\mathbf{x} \in S} \|\nabla_{\mathbf{x}} f(\boldsymbol{\theta}^t, \mathbf{x} + \boldsymbol{\varepsilon})\|$ |
| Adversarial alt. lower | at epoch $t$, i.e. $\boldsymbol{\theta}^t$ | $\mathbf{x} \in S$ | $\sup_{\mathbf{x} \in S} \frac{\|f(\theta^t, \mathbf{x}) - f(\theta^t, \mathbf{x} + \boldsymbol{\varepsilon})\|}{\|\boldsymbol{\varepsilon}\|}$ |

# S2    EXPERIMENTAL SETUP

This section contains comprehensive details on the experiments in the paper — this includes a summary of our strategy for Lipschitz upper bound calculation, models' architecture descriptions, hyperparameters and optimisation strategy choices for every experimental section in the main and supplementary parts of the paper. We also release our code on GitHub for further reproducibility and transparency.

In short, we benchmark our findings in a wide variety of settings, with different choices of (a) *architectures:* fully-connected networks (FCNs), convolutional neural networks (CNNs), Residual networks (ResNets) and Vision Transformers (ViT); (b) *datasets:* CIFAR-10 and CIFAR-100 (Krizhevsky, 2009), MNIST, MNIST1D (Greydanus, 2020) (a harder version of usual MNIST), as well as ImageNet; (c) *loss functions:* Cross-Entropy (CE) and Mean-Squared Error (MSE) loss. Unless stated otherwise, the results in the main paper are based on stochastic gradient descent with CE loss and the metrics have been averaged over 4 runs. All plots with shaded regions represent an

uncertainty of $\pm$ standard deviation from the mean, which is computed across different seeds. When semi-transparent dotted lines are shown, the solid line represents the mean over seeds and each dotted line depicts data from individual seeds.

## S2.1  LOCAL LIPSCHITZ ESTIMATES VS. PAIRS OF POINTS

As mentioned in Section 2, another way to lower bound is to consider Definition 2.1 and restrict the supremum computation to the train set. In essence, we want to compare the following lower bound estimates:

$$C_{\text{lower}} = \sup_{\mathbf{x} \in S} \|\nabla_{\mathbf{x}} f_{\boldsymbol{\theta}}\| \qquad \text{and} \qquad C_{\text{alt.lower}} = \sup_{\mathbf{x}, \mathbf{y} \in S, \mathbf{x} \neq \mathbf{y}} \frac{\|f_{\boldsymbol{\theta}}(\mathbf{x}) - f_{\boldsymbol{\theta}}(\mathbf{y})\|}{\|\mathbf{x} - \mathbf{y}\|}.$$

These bounds only converge when the considered set includes the whole function domain. Therefore it is not trivial to say which bound is better when a numerical estimation on a subset is performed.

To address this issue, we designed a simple experiment where we computed $C_{\text{lower}}$ and $C_{\text{alt.lower}}$ for an FCN ReLU network from Section 3.1 for several training epochs. The results of this experiment, displayed in Table S3, provide evidence for the fact that the $C_{\text{lower}}$ estimate is tighter than $C_{\text{alt.lower}}$.

Table S3: Comparison of two Lipschitz lower bound estimates. Computed for one seed.

| Epoch | $C_{\text{lower}}$ | $C_{\text{alt. lower}}$ |
|------:|------:|------:|
| 0 | 0.369 | 0.165 |
| 1 000 | 1.336 | 0.865 |
| 2 000 | 3.762 | 2.752 |
| 3 000 | 5.775 | 4.257 |
| 4 000 | 7.142 | 5.120 |
| 5 000 | 7.974 | 5.591 |
| 10 000 | 9.928 | 6.361 |
| 50 000 | 19.390 | 8.840 |

## S2.2  UPPER BOUND CALCULATION

We start by describing our approach to computing the upper bound of the Lipschitz constant, inspired by the AutoLip algorithm introduced by Virmaux & Scaman (2018). As briefly discussed in the main part of the paper (Section 2), we simply multiply per-layer Lipschitz bounds. For network $f_{\boldsymbol{\theta}}$ with $L$ layers, defined as $f_{\boldsymbol{\theta}} := f^{(L)} \circ \sigma \circ f^{(L-1)} \circ \sigma \circ \cdots \circ f^{(1)}$, where $\sigma$ is a 1-Lipschitz non-linear activation function, the upper bound looks as follows:

$$C \leq \prod_{i=1}^{L} \sup_{\mathbf{x}^{(i-1)} \in dom(f^{(i)})} \|\nabla_{\mathbf{x}^{(i-1)}} f^{(i)}\| \leq \prod_{i=1}^{L} \sup \|\nabla_{\mathbf{x}^{(i-1)}} f^{(i)}\| =: C_{\text{upper}} \qquad (2)$$

More pedantically, one can see this by using Theorem 2.1 and applying the chain rule:

$$C = \sup_{\mathbf{x} \in dom(f_{\boldsymbol{\theta}})} \|\nabla_{\mathbf{x}} f_{\boldsymbol{\theta}}\| = \sup_{\mathbf{x} \in dom(f_{\boldsymbol{\theta}})} \|\nabla_{f^{(L-1)}} f^{(L)} \cdot \nabla_{f^{(L-2)}} f^{(L-1)} \cdot \ldots \cdot \nabla_{f^{(1)}} f^{(2)} \cdot \nabla_{\mathbf{x}} f^{(1)}\|$$

$$\leq \sup_{f^{(L-1)}(\mathbf{x})} \|\nabla_{f^{(L-1)}} f^{(L)}\| \cdot \ldots \cdot \sup_{f^{(1)}(\mathbf{x})} \|\nabla_{f^{(1)}} f^{(2)}\| \cdot \sup_{\mathbf{x} \in dom(f_{\boldsymbol{\theta}})} \|\nabla_{\mathbf{x}} f^{(1)}\|$$

$$\leq \sup \|\nabla_{f^{(L-1)}} f^{(L)}\| \cdot \ldots \cdot \sup \|\nabla_{f^{(1)}} f^{(2)}\| \cdot \sup \|\nabla_{\mathbf{x}} f^{(1)}\| =: C_{\text{upper}}, \qquad (3)$$

where in the last line we consider the unconstrained supremum.

**Linear operations.** Each linear layer of the form $f^{(i)}(\mathbf{x}) = \mathbf{W}^{(i)} \mathbf{x}$ has Lipschitz constant $\|\mathbf{W}^{(i)}\|_2$, since the Jacobian of $f^{(i)}$ is simply the weight matrix. Convolutional layers are also linear operators and therefore can be similarly expressed as a linear transformation (considering that we perform proper flattening of the input image and reshaping in the end). According to Goodfellow

et al. (2016), equivalent linear transformations are represented by doubly block Toeplitz matrices which only depend on kernel weights. At the same time, Batch Normalisation layers are also per-feature linear transformations and, thus, the upper bound can be calculated as a maximum across per-feature Lipschitz constants.

**Activation functions.**   All activation functions that are used in this paper — ReLU, Leaky ReLU with slope $0.01$, max and average pooling — are also at most 1-Lipschitz and thus are considered 1-Lipschitz for the upper bound computation.

**Residual layers.**   Residual layers of the form $f(\mathbf{x}) = g(\mathbf{x}) + \mathbf{x}$, where $f, g : \mathbb{R}^n \to \mathbb{R}^n$ and $g$ is $C_g$-Lipschitz continuous, simply have a Lipschitz constant equal to $C_f = C_g + 1$. One can trivially derive this using the definition of Lipschitz continuity and the triangle inequality:

$$\begin{aligned}
\|f(\mathbf{x}) - f(\mathbf{y})\| &= \|g(\mathbf{x}) - g(\mathbf{y}) + \mathbf{y} - \mathbf{x}\| \\
&\leq \|g(\mathbf{x}) - g(\mathbf{y})\| + \|\mathbf{y} - \mathbf{x}\| && \text{(Triangle inequality)} \\
&\leq C_g \|\mathbf{x} - \mathbf{y}\| + \|\mathbf{x} - \mathbf{y}\| \\
&= (C_g + 1)\|\mathbf{x} - \mathbf{y}\| && (4)
\end{aligned}$$

**Attention layers.**   According to Kim et al. (2020), standard Attention layers are not Lipschitz continuous (or, in other words, have an unbounded Lipschitz constant) for unconstrained inputs. Therefore the upper bound introduced in Equation 2 is not applicable. Due to this reason, we only compute the lower Lipschitz bound in all experiments with Transformers.

*Remark on computational challenges.* Since equivalent linear weight matrices for convolutional operations assume flat input, the dimensions of this matrix grow rapidly with increasing image size and channel depth. This results in large memory consumption, which we leveraged by converting this matrix into the `scipy` sparse CSR format and using `scipy.sparse.linalg` library to compute the norm. We also found that standard `pytorch` implementation of 2-norm computation works rather slowly for large matrices. We have therefore implemented a Power method to compute 2-norms for linear layers and batched Jacobian matrices, resulting in almost $10\times$ speedup in some cases.

### S2.3   MODEL DESCRIPTIONS

#### S2.3.1   FCN ReLU NETWORKS

A fully Connected Network with ReLU activations (or FCN ReLU for short) consists of a sequence of Linear layers with zero bias term, each followed by a ReLU activation layer, the last layer included. When it is specified that FCN ReLU has a width of 256, there are only two linear transformations involved: first, from an input vector to a hidden vector of size 256, and second, from a hidden layer of size 256 to the output dimension. When a sequence of widths is given (i.e. FCN ReLU with widths 64,64), FCN has several hidden layers, sizes of which are listed from the hidden layer closer to the input to the hidden layer closer to the output.

*Remark on the Dead ReLU problem.* Since we use ReLU as the last layer's activation, outputs of the model can in some cases become zero-vectors, specifically in scenarios with classification using MSE loss. To address this issue we use a modified version of the FCN model for all MSE experiments — the last ReLU activation is substituted with a Leaky ReLU function with a negative slope of $0.01$.

Table S4 shows a list of FCN ReLU realisations with various hidden layer widths and depths and the respective number of model parameters. All models in this table are configured for the MNIST1D[S1] dataset.

#### S2.3.2   CNN NETWORKS

Our version of Convolutional Neural Networks (or CNNs for short) follows an approach similar to Nakkiran et al. (2020). We consider a 5-layer model, with 4 Conv-ReLU-MaxPool blocks, followed by a Linear layer with zero bias. All convolution layers have $3 \times 3$ kernels with stride=1, padding=1 and zero bias. Kernel output channels for convolution layers follow the pattern $[w, 2w, 4w, 8w]$,

---

[S1]MNIST1D input image is a vector of size 40.

Table S4: Table of the number of parameters of FCN ReLU models of various widths and depths, configured for the MNIST1D dataset.

| Model name | Number of parameters |
|---|---|
| FCN ReLU 16 | 800 |
| FCN ReLU 32 | 1,600 |
| ⋮ | ⋮ |
| FCN ReLU 80 | 4,000 |
| ⋮ | ⋮ |
| FCN ReLU 256 | 12,800 |
| ⋮ | ⋮ |
| FCN ReLU 800 | 40,000 |
| ⋮ | ⋮ |
| FCN ReLU 131072 | 6,553,600 |
| FCN ReLU 64,64 | 7,296 |
| FCN ReLU 64,64,64 | 11,392 |
| FCN ReLU 64,64,64,64 | 15,488 |
| FCN ReLU 64,64,64,64,64 | 19,584 |

where $w$ is the width of the model. Meanwhile, MaxPooling layers have kernels of sizes $[1, 2, 2, 8]$ for the case of CIFAR-10[S2] and $[1, 2, 2, 7]$ for the case of MNIST[S3]. This configuration shrinks the input image to a single vector that is then passed to the last Linear layer, yielding the output vector of size 10.

Table S5 displays a list of CNN realisations with various hidden layer widths and the respective number of model parameters. Since configurations for CIFAR-10 Krizhevsky (2009) and MNIST differ only in the MaxPooling layer size, the number of trainable parameters remains the same for both datasets.

Table S5: Table of the number of parameters of the CNN network of various widths.

| Model name | Number of parameters |
|---|---|
| CNN 5 | 9,985 |
| CNN 7 | 19,271 |
| CNN 10 | 38,870 |
| CNN 11 | 46,915 |
| CNN 12 | 55,716 |
| CNN 13 | 65,273 |
| ⋮ | ⋮ |
| CNN 20 | 153,340 |
| ⋮ | ⋮ |
| CNN 60 | 1,367,220 |

## S2.4 TRAINING STRATEGY

We present experiments that compare models that have a substantially varying number of parameters. To minimise the effect of variability in training, we painstakingly enforce the same learning rate, batch size and optimiser configuration for all models in one sweep. While this choice makes our

---

[S2]CIFAR-10 dataset input image has shape $32 \times 32 \times 3$.

[S3]MNIST dataset input image has shape $28 \times 28 \times 3$.

claims on the behaviour of the Lipschitz constant stronger, we now have the challenge of setting a reasonable stopping criterion to manage variable convergence rates.

For model $f_{\boldsymbol{\theta}}$ with parameter vector $\boldsymbol{\theta}$ and loss on the training set $\mathcal{L}(\boldsymbol{\theta}, S)$, after the end of each epoch we compute $\|\nabla_{\boldsymbol{\theta}}\mathcal{L}(\boldsymbol{\theta}, S)\|_2$, which we call gradient norm for simplicity. In all experiments, unless stated otherwise, we control model training by monitoring the respective gradient norm — if it reaches a small value (ideally zero), our model has negligible parameter change (i.e. $\|\boldsymbol{\theta}^{t+1} - \boldsymbol{\theta}^t\|_2$ is small) or, in other words, has reached a local minimum. By means of experimentation, we found that stopping models at 0.01 gradient norm value gives good results for most scenarios.

*Possible pitfalls.* In most cases, during the course of training, the gradient norm is at first relatively small (in the range of $10^{-4}$ to $10^{-2}$), then rapidly increases and then slowly decreases. If our only stopping criteria is minuscule gradient norm, we may end up early stopping models right after a few epochs. To avoid this we also introduce a minimum number of epochs that each model has to train for.

## S2.5 LEARNING RATE SCHEDULERS

This section thoroughly describes learning rate schedulers (LR schedulers for short) that we use in our experiments. Each scheduler modifies a variable $\gamma_t$, which is a coefficient that is multiplied by some base learning rate (i.e. learning rate at time step $t$ is $\eta_t = \gamma_t \cdot \eta$, where $\eta$ is the base learning rate). Note that we perform scheduler updates at every parameter update (which happens several times per epoch), and the schedulers are aware of the dataset length and batch size to adapt to epoch-based settings accordingly.

**Warmup20000Step25 LR Scheduler (Figure S1a).** This scheduler linearly scales the learning rate from a factor of $\frac{1}{20,000}$ to 1 for 20,000 updates. Next, the scaling coefficient drops by a factor of 0.75 every 25% of the next 10,000 epochs. Afterwards, the coefficient remains at the constant factor of $0.75^3 = 0.421875$.

**Cont100 LR Scheduler (Figure S1b).** This scheduler continuously drops the scaling coefficient by a factor of 0.95 every 100 epochs.

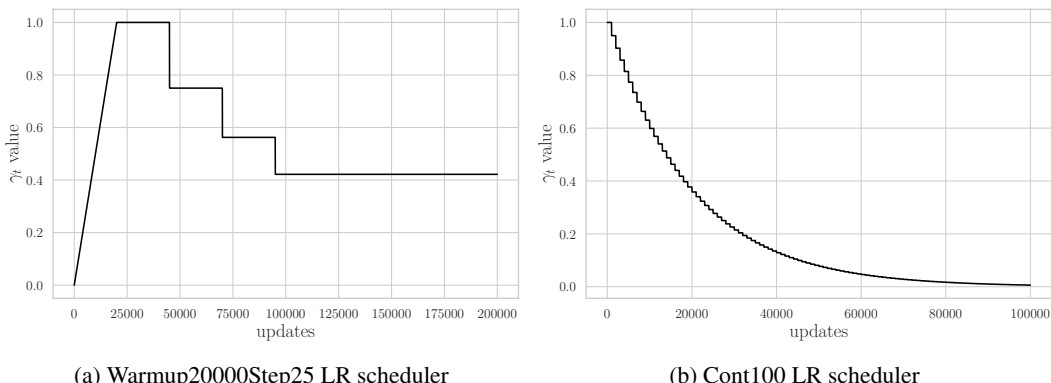

(a) Warmup20000Step25 LR scheduler          (b) Cont100 LR scheduler

Figure S1: Learning rate scaling coefficient $\gamma_t$ against updates for two different learning rate schedulers. In each case there are 10 updates per epoch.

## S2.6 DESCRIPTION OF EXPERIMENTAL SETTINGS

This section presents a detailed description of each experimental setup. We also include theoretical Double Descent interpolation thresholds for Double Descent experiments, where in each equation $n$ stands for the number of training samples, $K$ for the number of classes and $p$ for the number of parameters in the model.

### S2.6.1 A SIMPLE TOY EXAMPLE TO DEMONSTRATE THE FIDELITY OF THE LOWER LIPSCHITZ BOUND (FIGURE 5)

For this experiment, we set our data domain to be $\mathcal{D} = [-5, 5]^2$, and the three equally spaced multinomial Gaussians. Each Gaussian's mean is $1.5$ units away from the origin and has $\Sigma = I_2$. We sampled 15 points from each Gaussian, resulting in a dataset of 45 points. The dataset with the true labels is depicted in Figure S2.

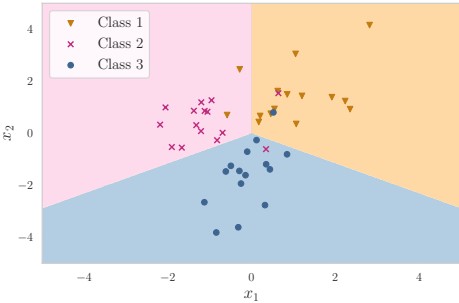

Figure S2: The dataset and the true labels.

For the classifier, we chose an FCN ReLU Network with 2 hidden layers of width 100. We optimised the model on Cross-Entropy loss with Gradient Descent and a constant learning rate of $0.02$ for $100,000$ epochs. To compute the Lipschitz bounds, we sampled $1,000,000$ points on the $[-5, 5]^2$ grid. Additionally, we also computed the Jacobian norms outside of the data domain, i.e. on the $[-10, 10]^2$ grid, which resulted in the same estimate of $144.194$. The results for the local Lipschitz computation are shown in Figure S3.

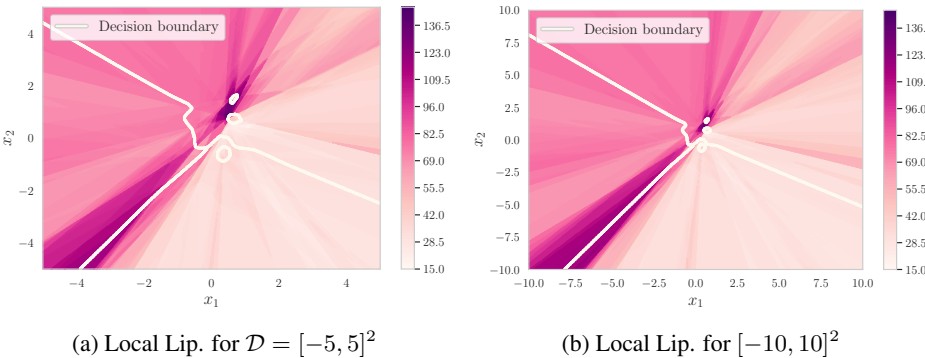

(a) Local Lip. for $\mathcal{D} = [-5, 5]^2$       (b) Local Lip. for $[-10, 10]^2$

Figure S3: Local Lipschitz constants of $f$ for various input domains.

### S2.6.2 DOUBLE DESCENT ON MNIST1D, FCN RELU NETWORKS, CROSS-ENTROPY LOSS (FIGURE S12)

For this experiment, we trained a sweep of FCN ReLU models (see S2.3.1) with widths [16, 32, 64, 80, 96, 128, 256, 512, 1024, 2048, 4096, 8192, 16384, 32768, 65536, 131072] on MNIST1D[S4] with batch size 512 using Cross-Entropy loss and SGD optimiser without momentum. We used a Warmup20000Step25 LR scheduler (see S2.5) and a base learning rate of 0.005. We trained our models for at least 10,000 epochs and stopped each model when either 0.01 gradient norm is reached or when 300,000 epochs have passed. We trained 4 seeds for each run. The theoretical threshold for this scenario is at $n \approx p$, which corresponds to FCN ReLU 80 (4000 parameters).

*Comment on the hyperparameter choice.* In this experiment, we used a very small learning rate to smoothly fit both under- and over-parametrised models. Therefore we require a significant amount of training epochs to secure convergence for all settings.

*Comment on Figure S12.* In the figure, the training loss uncertainty for models from width 8192 to 65536 is lower bounded by zero (since training loss cannot be negative) and therefore is depicted as a vertical line in the log-log plot.

### S2.6.3 DOUBLE DESCENT ON MNIST1D, FCN RELU NETWORKS, MSE LOSS (FIGURE S13)

This experiment depicts a sweep of FCN ReLU models (see S2.3.1) with widths [16, 32, 64, 80, 96, 128, 256, 512, 1024, 2048, 4096, 8192, 16384, 32768, 65536], trained on MNIST1D[S4] with batch size 512 using MSE loss and SGD optimiser without momentum. We used a Warmup20000Step25 LR scheduler (see S2.5) and a base learning rate of 0.001. We trained our models for at least 10,000 epochs and stopped each model when 0.01 gradient norm is reached or when 200,000 epochs have passed. We trained 4 seeds for each run. The theoretical threshold for this scenario is at $n \cdot K \approx p$, which corresponds to FCN ReLU 800 (40,000 parameters).

### S2.6.4 DOUBLE DESCENT ON CIFAR-10, CNN NETWORKS, CROSS-ENTROPY LOSS (FIGURE S14)

For this experiment, we trained a sweep of CNN models with widths [5, 7, 10, 11, 12, 15, 20, 25, 30, 35, 40, 45, 50, 55, 60] on CIFAR-10[S5] with batch size 128 using Cross-Entropy loss and SGD optimiser without momentum. We used a Cont100 LR scheduler (see S2.5) and a base learning rate of 0.01. We trained our models for at least 500 epochs and stopped each model when 0.01 gradient norm is reached or when 5000 epochs have passed. We trained 4 seeds for each run. The theoretical threshold for this scenario is at $n \approx p$, which corresponds to somewhere between CNN 11 (46,915 parameters) and CNN 12 (55,716 parameters).

### S2.6.5 DOUBLE DESCENT ON CIFAR-100 WITH 20 SUPERCLASSES, CNN NETWORKS, CROSS-ENTROPY LOSS (FIGURE 7)

For this experiment, we trained a sweep of CNN models with widths [5, 7, 10, 11, 12, 13, 15, 20, 25, 30, 35, 40, 45, 50, 55, 60] on CIFAR-100[S6] with 20 superclasses, as described by Krizhevsky (2009) with batch size 128 using Cross-Entropy loss and SGD optimiser without momentum. We used a Cont100 LR scheduler (see S2.5) and a base learning rate of 0.005. We trained our models for at least 500 epochs and stopped each model when 0.01 gradient norm is reached or when 15 000 epochs have passed. We trained 4 seeds for each run. The theoretical threshold for this scenario is at $n \approx p$, which corresponds to somewhere between CNN 11 (47 795 parameters) and CNN 12 (56 676 parameters).

### S2.6.6 DOUBLE DESCENT ON MNIST, CNN NETWORKS, CROSS-ENTROPY LOSS (FIGURE S15)

For this experiment, we trained a sweep of CNN models with widths [5, 7, 10, 11, 12, 13, 15, 20, 25, 30, 35, 40, 45, 50, 55, 60] on MNIST[S7] with a fixed 10% label shuffling (i.e. the dataset is the

---

[S4]MNIST1D has 4000 training samples.

[S5]CIFAR-10 has 50,000 training samples.

[S6]CIFAR-100 has 50 000 training samples.

[S7]MNIST has 60,000 training samples.

same across all seeds and models) with batch size 128 using Cross-Entropy loss and SGD optimiser without momentum. We used a Cont100 LR scheduler (see S2.5) and a base learning rate of 0.01. We trained our models for at least 100 epochs and stopped each model when 0.01 gradient norm is reached or when 1000 epochs have passed. We train 4 seeds for each run. The theoretical threshold for this scenario is at $n \approx p$, which corresponds to somewhere between CNN 12 (55,500 parameters) and CNN 13 (65,039 parameters). Note that the parameter count is different due to different input size.

### S2.6.7 DOUBLE DESCENT ON CIFAR-10, VISION TRANSFORMER (VIT) NETWORKS (FIGURE S16)

For this experiment, we trained a sweep of ViT models with widths [3, 5, 7, 8, 10, 11, 12, 13, 14, 15, 16, 17, 18, 19, 20, 50, 100, 500]. The width is used as the last dimension of output tensor after linear transformation, as well the dimension of the FeedForward layer (we multiply it by 4 in this case). No dropout is used. The patch size is equal to 8, and we use 6 heads with 6 Transformer blocks. We used the `vit-pytorch` Python package implementation.

All models were trained on CIFAR-10[S8] with batch size 128 using Cross-Entropy loss and SGD optimiser without momentum. We used a Step LR scheduler (see S2.5) and a base learning rate of 0.05. We trained our models for at least 2,000 epochs and stopped each model when 0.01 gradient norm is reached or when 10,000 epochs have passed. We train 4 seeds for each run. The theoretical threshold for this scenario is at $n \approx p$, which corresponds to somewhere near ViT 5 (49,099 parameters).

*Remark.* Since the Attention layers are not Lipschitz continuous (Kim et al., 2020), or, rather, the upper bound on the Lipschitz constant does not exist in $\mathbb{R}^d$, we only present the results for the lower Lipschitz bounds. In order to provide proper upper bounds, modification to Attention layers are required, as described in detail by Kim et al. (2020).

### S2.6.8 LIPSCHITZ BOUNDS EVOLUTION, FCN RELU, MNIST1D WITH CONVEX COMBINATIONS (FIG. 1)

To showcase Lipschitz constant evolution we trained 4 FCN ReLU 256 with different seeds on the MNIST1D dataset with batch size 512 for 83,000 epochs. Models were trained using Cross-Entropy loss and SGD optimiser with a base learning rate of 0.005 and Warmup20000Step25 LR scheduler (see S2.5). Each model achieved gradient norm of 0.01 up to 2 significant figures. Each epoch consists of 8 parameter updates.

For this scenario, we empirically defined the stable phase to begin from epoch 2500 (or after 20,000 updates). The slopes for the upper, lower, and average Lipschitz bounds are: 0.59, 0.46 and 0.44 respectively and are computed by examining the slope coefficient of a linear regression model fitted to the corresponding values in the log-log scale. The $R^2$ values of the fit for the upper, lower, and average Lipschitz bounds are 0.9955, 0.9986 and 0.9980 respectively.

To compute the lower bound on convex combinations of samples from MNIST1D we constructed a set $S^*$, which contains: (a) training set $S$ — 4000 samples, (b) test set $S'$ — 1000 samples, (c) convex combinations $\lambda \mathbf{x}_i + (1 - \lambda)\mathbf{x}_j$ from $S$ — 100,000 samples for each $\lambda = \{0.1, 0.2, 0.3, 0.4, 0.5\}$, and (d) convex combinations $\lambda \mathbf{x}_i + (1 - \lambda)\mathbf{x}_j$ from $S'$ — 100,000 samples for each $\lambda = \{0.1, 0.2, 0.3, 0.4, 0.5\}$. Altogether this makes $S^*$ contain 1,005,000 samples.

### S2.6.9 LIPSCHITZ BOUNDS EVOLUTION, RESNET50, SUBSET OF 200,000 IMAGENET SAMPLES (FIGURE 2)

For this experiment, we trained 3 ResNet50 models with different seeds for 90 epochs on full ImageNet with batch size 256, using Cross-Entropy loss and SGD optimiser with momentum of 0.9, weight decay of 0.0001 and base learning rate 0.1. We also used a LR decay scheme where the learning rate is decreased 10 times every 30 epochs. We then evaluated lower Lipschitz bounds for epochs [0, 1, 10, 20, 30, 40, 50, 60, 70, 80, 90] on a fixed random subset of 200,000 images from the ImageNet training set. During Lipschitz evaluation, all training samples are resized to $256 \times 256 \times 3$, then center-cropped to size $224 \times 224 \times 3$ and then normalised using $mean = [0.485, 0.456, 0.406]$

---

[S8]CIFAR-10 has 50,000 training samples.

and $std = [0.229, 0.224, 0.225]$. During training, training samples are randomly resized and cropped to $224 \times 224 \times 3$ and then normalised using the same $mean$ and $std$.

For this scenario, we empirically defined the stable phase to begin from epoch 60. The slopes for the upper, lower, and average Lipschitz bounds are: 7.32, 0.49 and 0.48 respectively and are computed by examining the slope coefficient of a linear regression model fitted to the corresponding values in the log-log scale. The $R^2$ values of the fit for the upper, lower, and average Lipschitz bounds are 0.8441, 0.9718 and 0.8774 respectively.

### S2.6.10  DISTRIBUTION OF THE NORM OF THE PER-SAMPLE JACOBIAN FOR RESNET18 TRAINED ON IMAGENET (FIGURE S8)

For this experiment we first evaluated the norms of the Jacobian matrices (see 2.1) of ResNet18 for each sample of ImageNet. We took a pretrained ResNet18 from `pytorch hub`. We then constructed another dataset, which took a mean of all possible pairs of 1,000 ImageNet samples with the highest Jacobian norm from the previous calculation, and evaluated the distribution once more on the new dataset.

### S2.6.11  VARIANCE UPPER BOUNDS AND BIAS-VARIANCE TRADEOFF (SECTIONS 4 AND S3.11)

For this study we used the same models that we have trained for the Double Descent on MNIST1D using FCN ReLU networks trained with MSE setting (see S2.6.3).

To compute variance bound estimates (see Eq. Var bound 1) in Figure 8, all expectations and variances are computed as their respective unbiased statistical estimates over 4 seeds. $\overline{C}$ is estimated as the norm of the average Jacobian among 4 seeds on the training set:

$$\overline{C} = \sup_{\mathbf{x} \in \mathcal{D}} \|\nabla_{\mathbf{x}} \overline{f_{\boldsymbol{\theta}}(\mathbf{x})}\| = \sup_{\mathbf{x} \in \mathcal{D}} \|\nabla_{\mathbf{x}} \mathbb{E}_{\zeta}[f_{\boldsymbol{\theta}}(\mathbf{x}, \zeta)]\| = \sup_{\mathbf{x} \in \mathcal{D}} \|\mathbb{E}_{\zeta}[\nabla_{\mathbf{x}} f_{\boldsymbol{\theta}}(\mathbf{x}, \zeta)]\|$$

$$\geq \sup_{\mathbf{x} \in S} \|\mathbb{E}_{\zeta}[\nabla_{\mathbf{x}} f_{\boldsymbol{\theta}}(\mathbf{x}, \zeta)]\| \approx \sup_{\mathbf{x} \in S} \|\frac{1}{4} \sum_{i=1}^{4} (\nabla_{\mathbf{x}} f_{\boldsymbol{\theta}}(\mathbf{x}, \zeta_i))\| \tag{5}$$

### S2.6.12  EFFECT OF THE LOSS FUNCTION ON THE LIPSCHITZ CONSTANT (SECTION S3.17)

For this study we used the same models that we have trained for the Double Descent on MNIST1D using FCN ReLU networks trained with Cross-Entropy setting (see S2.6.2) and with MSE setting (see S2.6.3). We plotted the evolution for the first 83,000 epochs for all models.

### S2.6.13  EFFECT OF THE OPTIMISATION ALGORITHM ON THE LIPSCHITZ CONSTANT (SECTION S3.18)

For this study, we used the same models that we have trained for the Double Descent on MNIST1D using FCN ReLU networks trained with Cross-Entropy setting (see S2.6.2), as well as a set of 4 additionally trained FCN ReLU models with various seeds that were optimised with standard `pytorch` Adam optimiser ($\beta_1 = 0.9, \beta_2 = 0.999$). Each of these models was trained on MNIST1D with Cross-Entropy loss, 0.005 learning rate and Warmup20000Step25 LR scheduler (see S2.5). Parameter vector was computed as a concatenation of flattened layer weights at each layer.

In Figure S29a we showcase models up to epoch 10,000, while in figure S29b models are stopped at 83,000 and 1,100 epochs for the SGD and Adam case respectively. In the former plot, both models achieved a gradient norm of at most 0.01 up to 2 significant figures at the end of their training.

*Comment on Figure S29a.* We display training up to 10,000 epochs even though Adam reached low gradient norm much earlier due to slower rate of SGD — by 1,100 epochs SGD is still in the early training phase.

### S2.6.14  EFFECT OF DEPTH OF THE NETWORK ON ITS LIPSCHITZ CONSTANT (SECTION S3.19)

For this experiment we trained 5 types of FCN ReLU models with increasing depth. In particualr, we considered FCN ReLU 64; FCN ReLU 64,64; FCN ReLU 64,64,64; FCN ReLU 64,64,64,64 and

FCN ReLU 64,64,64,64,64. Each model was trained on MNIST1D with batch size 512 using Cross-Entropy loss and the SGD optimiser, 0.005 learning rate and Warmup20000Step25 LR scheduler (see S2.5). We trained our models for at least 10,000 epochs and stopped each model when either 0.01 gradient norm is reached or when 300,000 epochs have passed. We trained 4 seeds for each run.

For this scenario, we computed slopes for Lipschitz bounds starting from depth 2. The slopes for the upper, lower, and average Lipschitz bounds for *trained networks* are: 3.33, 2.03, 1.19 respectively and are computed by examining the slope coefficient of a linear regression model fitted to the corresponding values in the log-log scale. The $R^2$ values of the fit for the upper, lower, and average Lipschitz bounds are: 0.9749, 0.9483 and 0.8798 respectively. For networks *at initialisation*, slopes from depth 2 for the upper, lower, and average Lipschitz bounds are 0.30, -2.52 and -2.84 respectively.

### S2.6.15   EFFECT OF THE NUMBER OF TRAINING SAMPLES ON THE LIPSCHITZ CONSTANT (SECTION S3.20)

For this experiment we trained 4 types of FCN ReLU 256 models on different sizes of MNIST1D: 4000, 1000, 500 and 100 training samples. Sampling was performed by taking a random subsample from the main dataset. Each model was trained with batch size 512 using Cross-Entropy loss and the SGD optimiser, 0.005 learning rate and Warmup20000Step25 LR scheduler (see S2.5). We trained our models for at least 10,000 epochs and stopped each model when either 0.01 gradient norm is reached or when 300,000 epochs have passed. We trained 4 seeds for each run.

### S2.6.16   EFFECT OF SHUFFLING LABELS (SECTION 5)

For this experiment, we trained a sweep of CNN models with widths [5, 7, 10, 11, 12, 13, 14, 15, 16, 17, 18, 19, 20, 30, 40] on CIFAR-10 with various amounts of shuffled labels ($\alpha$ parameter): 0%, 10%, 20%, 30%, 40%, 50%, 60%, 70%, 80%, 90% and 100%. Each shuffle is incremental, meaning that all shuffles from a dataset with a smaller $\alpha$ are contained in the dataset with a larger $\alpha$. For each dataset, subsets and shuffles are fixed among seeds. Each model was trained with batch size 128 using Cross-Entropy loss and the SGD optimiser, 0.01 learning rate and Cont100 LR scheduler (see S2.5), where the minimum learning rate was limited to $0.01 \cdot 0.001 = 1\text{e-}5$. We trained the models for at least 1 000 epochs and stopped each model when either 0.01 gradient norm is reached or when 20 000 epochs have passed.

*Comment on the minimum number of epochs.* In comparison to the similar Double Descent setup (see S2.6.4) we use a larger number of minimum epochs in this experiment, since smaller models require more updates to escape the initialisation well, especially in the presence of label noise (see S2.4).

### S2.6.17   EFFECT OF DROPOUT AND WEIGHT DECAY (SECTIONS S3.21 AND S3.23)

For the following set of experiments, we used the approach of Idelbayev, who carefully implements the approach by He et al. (2016). We refer to the original paper and the code repository cited before for implementation details. The only difference in our experimental setup is the inclusion of dropout layers (for the weight decay scenario, we do not use dropout). A 2D version of the Dropout layer is inserted before each ResNet block (i.e. a group of layers with a skip connection), leaving the first Convolution layer and BatchNorm intact. We also add a 1D Dropout layer before the last linear map. All dropout layers use the same probability $p$. We use the following values of $p$ for different runs: $[0.0, 0.1, 0.2, 0.25, 0.3, 0.4]$. For all Dropout experiments, we use the default value of 1e-4 for weight decay and train all models for 500 epochs.

For the weight decay experiments, we use the original setup and only change the weight decay parameter. We used the following set of values: [0.0, 1e-4, 2e-4, 3e-4, 4e-4, 5e-4, 1e-3, 2.5e-3]. All models were trained for 500 epochs.

## S3 ADDITIONAL EXPERIMENTS

### S3.1 LINEARISATION OF NEURAL NETWORKS AND THE EFFECT OF TRAINING

Using the first order of the Taylor expansion on our Neural Network $f(\mathbf{x}; \boldsymbol{\theta})$, we get the following linearisation: $f(\mathbf{x}; \boldsymbol{\theta}) \approx f(\mathbf{x}; \boldsymbol{\theta}_0) + \langle \boldsymbol{\theta} - \boldsymbol{\theta}_0, \nabla_{\boldsymbol{\theta}} f(\mathbf{x}; \boldsymbol{\theta}_0) \rangle$. To see the effects of training on the lower Lipschitz constant bound, we can take the derivative w.r.t. input of the expression above and compute the 2-norm:

$$\sup_{\mathbf{x} \in S} \|\nabla_{\mathbf{x}} f(\mathbf{x}; \boldsymbol{\theta})\| \approx \sup_{\mathbf{x} \in S} \|\nabla_{\mathbf{x}} f(\mathbf{x}; \boldsymbol{\theta}_0) + \langle \boldsymbol{\theta} - \boldsymbol{\theta}_0, \frac{\partial^2}{\partial \mathbf{x} \partial \boldsymbol{\theta}} f(\mathbf{x}; \boldsymbol{\theta}_0) \rangle \|$$

$$\leq \sup_{\mathbf{x} \in S} \|\nabla_{\mathbf{x}} f(\mathbf{x}; \boldsymbol{\theta}_0)\| + \|\boldsymbol{\theta} - \boldsymbol{\theta}_0\| \cdot \sup_{\mathbf{x} \in S} \|\frac{\partial^2}{\partial \mathbf{x} \partial \boldsymbol{\theta}} f(\mathbf{x}; \boldsymbol{\theta}_0)\| .$$

We compute the bound above for an FCN ReLU 256 network trained on MNIST1D with CE loss (see Appendix S2.6.2). It is worth noting that even in this experiment with a small network (12,800 params) and a small dataset (4,000 train points wih 10 classes, input dimension is 40), computing the second derivative becomes very expensive, as we have to evaluate a $10 \times 12,800 \times 40$ tensor for all 4,000 train points (that is more than half a billion gradient evaluations). We therefore restricted our computation to a subset of 470 training points that we managed to compute given our time constraints. We compute the norm of this tensor in its matrix representation, where we flatten the output-parameter dimension (for this example, we have a $128,000 \times 40$ matrix).

The results are as follows: the lower bound at the last epoch (LHS) is $24.228$, while the lower bound at initialisation is almost 65 times smaller: $0.369$. The second derivative term turns out to be only $4.685$, which shows that the final Lipschitz constant changes significantly during training (due to large parameter vector change).

*Remark.* The mean $\pm$ std. of $\|\frac{\partial^2}{\partial \mathbf{x} \partial \boldsymbol{\theta}} f(\mathbf{x}; \boldsymbol{\theta}_0)\|$ is $3.127 \pm 0.619$, showing that the estimate is very similar for most training points. This behaviour is expected, as we the network has not yet trained and is smooth around the whole domain due to the random initialisation.

### S3.2 USING ADVERSARIALLY PERTURBED SAMPLES FOR THE LOWER LIPSCHITZ BOUND COMPUTATION

One way to improve the lower Lispchitz bound estimate (see Equation 1) is to consider adversarial perturbations of the input instead of the inputs themselves. For this experiment, we perturb input samples using a slight modification of a PGD attack (He et al., 2016; Kurakin et al., 2017) with $\epsilon = 0.5$, where the learning rate of 10.0 is decayed by a factor of 0.95 for each step in the algorithm. We use 1,000 iterations of PGD for each sample. We also compute the following Lipschitz estimates:

$$C_{\text{lower}, \epsilon=0} := \sup_{\mathbf{x} \in S} \|\nabla_{\mathbf{x}} f_{\boldsymbol{\theta}}(\mathbf{x})\|_2 ,$$

$$C_{\text{lower}, \epsilon=0.5} := \sup_{\mathbf{x} \in S} \|\nabla_{\mathbf{x}} f_{\boldsymbol{\theta}}(\mathbf{x} + \boldsymbol{\varepsilon})\|_2 ,$$

$$C_{\text{adv.straightforward}, \epsilon=0.5} := \sup_{\mathbf{x} \in S} \frac{\|f_{\boldsymbol{\theta}}(\mathbf{x} + \boldsymbol{\varepsilon}) - f_{\boldsymbol{\theta}}(\mathbf{x})\|_2}{\|\boldsymbol{\varepsilon}\|_2} ,$$

where $\boldsymbol{\varepsilon}$ is a noise vector, generated by the PGD attack ($\|\boldsymbol{\varepsilon}\|_2 \leq \epsilon$).

Table S6 shows the computed bounds for CNN models from the Double Descent experiment on CIFAR-10 (see Appendix S2.6.4) on one fixed seed. The results clearly show that adversarial perturbations do not always result in tighter lower bounds, and if they do, the increase in the estimate is not substantial.

A similar behaviour can be observed for other values of $\epsilon$. Figure S4 shows how close the $C_{\text{lower}}$ bounds lie for different attack strengths, while the loss on perturbed test values increases significantly.

### S3.3 EVALUATING LIPSCHITZ BOUNDS FOR OUT-OF-DISTRIBUTION (OOD) SAMPLES

To study the effects of out-of-distribution (OOD) samples on the Lipschitz constant, we took the models from the MNSIT1D, CE Double Descent study (Appendix S2.6.2) and computed the Lipschitz

Table S6: Lipschitz bounds for various CNN models, trained on CIFAR-10. Adversarial examples were generated using a PGD attack with $\epsilon = 0.5$.

| | CNN 5 | CNN 7 | CNN 10 | CNN 20 | CNN 30 | CNN 40 |
|---|---|---|---|---|---|---|
| $C_{\text{lower},\epsilon=0}$ | **148.29** | 633.68 | **756.37** | 172.15 | 123.61 | **106.35** |
| $C_{\text{lower},\epsilon=0.5}$ | 139.54 | **658.03** | 739.00 | **173.07** | **129.13** | 102.14 |
| $C_{\text{adv.straightforward},\epsilon=0.5}$ | 118.21 | 507.00 | 601.16 | 140.54 | 102.74 | 86.84 |

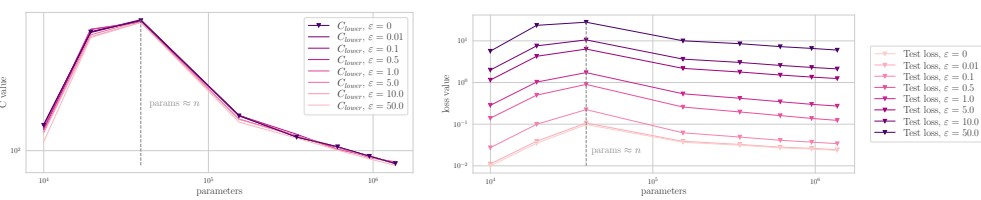

(a) Lip. bounds on perturbed train samples    (b) Loss on perturbed test samples

Figure S4: Comparison of various Lipschitz constant bounds with test loss with increasing hidden layer width, in the case of CNN networks on perturbed CIFAR-10 samples.

constant bounds on OOD train samples, as well as test loss on OOD test samples. To generate an OOD sample, we added standard Gaussian noise with scale $r$ to the sample. The results are shown in Figure S5, which confirm that lower Lipschitz bounds do not change significantly when evaluated at OOD points.

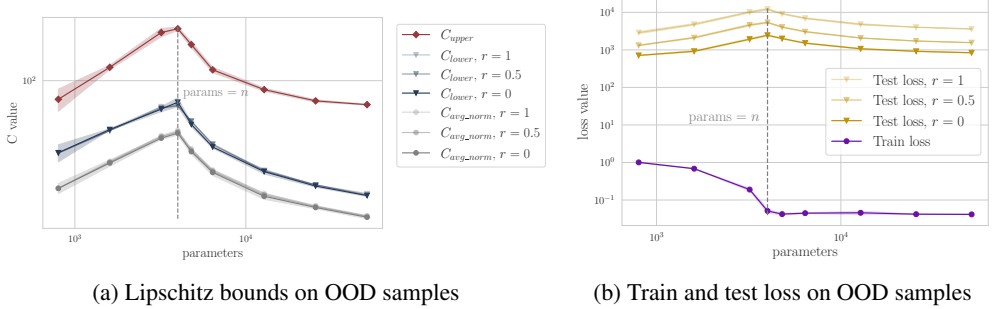

(a) Lipschitz bounds on OOD samples    (b) Train and test loss on OOD samples

Figure S5: Comparison of various Lipschitz constant bounds with test loss with increasing hidden layer width, in the case of FCN ReLU networks on OOD MNIST1D samples.

### S3.4 EVALUATING THE TIGHTNESS OF THE THEORETICAL BOUND

To compare the bounds derived in Appendix S4.1, we trained one FCN ReLU model with 6 hidden layers (widths 100, 100, 80, 80, 60 and 60) on MNIST1D with SGD using batch size 1,000 (4 updates per epoch). We also used a learning rate of 0.04 and a Cont50 LR scheduler (similar to Cont100, but the decay applies every 50 epochs, or every 200 updates in this case).

Figure S6 shows the results of the evaluation. We denote Bound 2 and Bound 1 as follows:

$$\text{Bound 1}(\tau) = \frac{2}{r} C_{\boldsymbol{\theta}}^{\text{discrete}}(\tau) \sum_{t=0}^{\tau-1} \|\boldsymbol{\theta}^{t+1} - \boldsymbol{\theta}^t\|_2 + C_{\mathbf{x}}(\boldsymbol{\theta}^0),$$

$$\text{Bound 2}(\tau) = \frac{2}{r} C_{\boldsymbol{\theta}}^{\text{discrete}}(\tau) B \eta \tau + C_{\mathbf{x}}(\boldsymbol{\theta}^0),$$

where $C_{\boldsymbol{\theta}}^{\text{discrete}}(\tau) = \sup_{\boldsymbol{\theta} \in \{\boldsymbol{\theta}^0, \ldots, \boldsymbol{\theta}^\tau\}} \sup_{\mathbf{x} \in S} \|\nabla_{\boldsymbol{\theta}} f(\boldsymbol{\theta}, \mathbf{x})\|$ denotes the Lipschitz constant with respect to model parameters up to checkpoint $\tau$, estimated at the train samples.

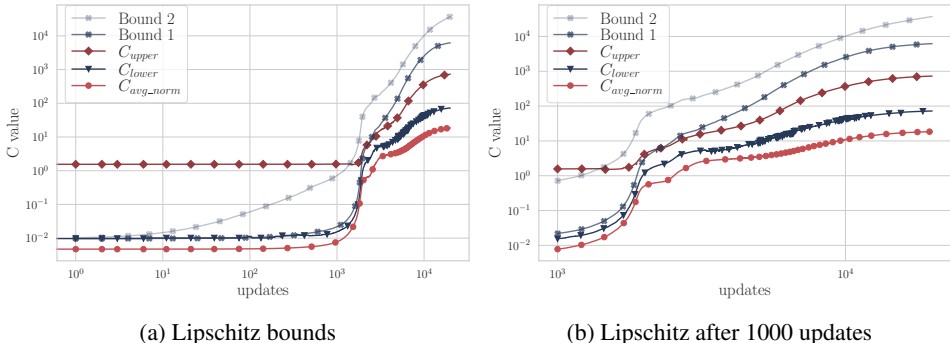

(a) Lipschitz bounds             (b) Lipschitz after 1000 updates

Figure S6: Evaluation of the theoretical bounds on the Lipschitz evolution and comparison with the actual upper and lower Lipschitz estimates. Computed for FCN ReLU 100,100,80,80,60,60 trained on MNIST1D with SGD with batch size 1,000 (4 updates per epoch).

The results show that our both bounds follow a trend similar to the other Lipschitz estimates. Moreover, at the beginning of training, our bound comes out to be smaller than the upper bound estimate.

### S3.5 LIPSCHITZ EVOLUTION OF A VISION TRANSFORMER ON A SUBSET OF IMAGENET

In this experiment, I evaluate the evolution of lower Lipschitz bounds for the Vision Transformer on a subset of ImageNet. As previously discussed, Attention layers do not have a theoretical Lipschitz upper bound and hence I only present the lower and average Lipschitz bounds. For this experiment, a subset of 50 000 train points from ImageNet was taken and only one seed is considered due to computational and time limitations. I used a `SimpleViT` model from the `vit_pytorch` package, where the patch size is set to 16, the last dimension after linear transformation is 384, depth is set 12, number of heads to 6, and the MLP dimension to 1 536. The model was pretrained on full ImageNet. Figure S7 shows the results, with a comparison to ResNet50 bound on the same 50 000 sample subset.

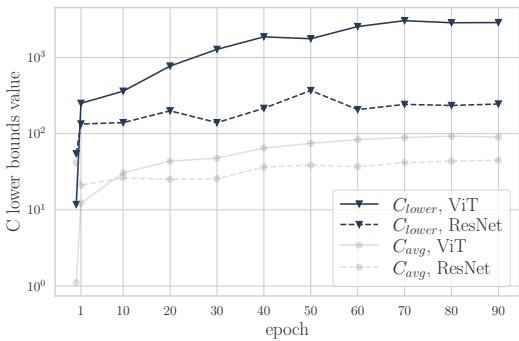

Figure S7: Lower Lipschitz bounds evolution for ViT and ResNet50 on a 50 000 samples ImageNet subset.

### S3.6 ESTIMATION OF THE POTENTIAL ERROR OF EVALUATING LOWER LIPSCHITZ BOUND BY CONSIDERING A SUBSAMPLE OF IMAGENET

Due to computational limitations, in the ResNet50 evolution experiment (see Section 3.2) we evaluate the lower Lipschitz bound on a subset of ImageNet. This decision would naturally produce a weaker estimate of the Lipschitz constant, compared to the full dataset evaluation, raising concerns on the representativeness of the lower bound in the first place. Therefore we conducted a small investigation into the magnitude of potential error due to ImageNet subsampling for the ResNet18 case.

In this analysis, we took the distribution of Jacobian norms for pretrained ResNet18, evaluated for the whole ImageNet training dataset (see S2.6.10 for more details), and estimated the lower bound for various random subsets of computed norms. Results are depicted in Table S7. According to the obtained statistics, doubling the subset size decreases the relative error by around 5%, suggesting that moderate errors can be achieved for rather small subsets. These results motivated our decision of using 200,000 as our subsample size — a plausible trade-off between computational burden and estimation accuracy.

Table S7: Estimates of the lower bound for trained ResNet18 on the different subsets of ImageNet. Estimates are averaged over 1000 random subsets.

| Subset size | Estimate | St.dev. | Percentage difference to true estimate |
|---|---|---|---|
| 50,000 | 220.50 | 23.72 | 20.87% |
| 100,000 | 234.95 | 23.49 | 15.69% |
| 200,000 | 249.28 | 21.24 | 10.54% |
| 300,000 | 257.02 | 18.85 | 7.76% |
| 400,000 | 262.15 | 16.86 | 5.92% |

### S3.7 THE CONVEX COMBINATION STUDY FOR RESNET18

In this experiment, we take the top $1,000$ samples that have the highest Jacobian norm, consider their convex combinations, and use that as a basis to evaluate the lower Lipschitz bounds. In fact, in Figure S8, we show the entire distribution of norms for these "hard" convex combinations as well as the entire ImageNet training set.

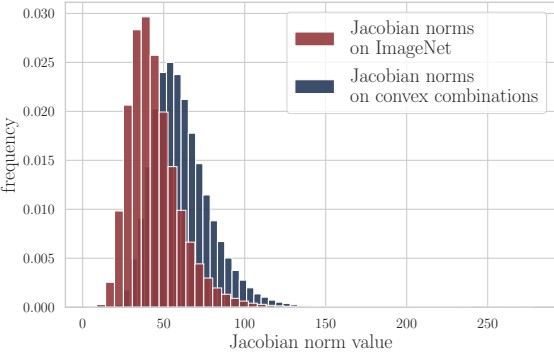

Figure S8: Distribution of the norm of the per-sample Jacobian for pretrained **ResNet18**, computed on the entire ImageNet and 1,000,000 hard convex combinations on ImageNet (See Appendix S2.6.10 for more).

We find that while the distribution indeed shifts towards larger per-sample Jacobian norms for the hard convex combinations, the shift is not even a multiplicative factor of $2\times$ more. This shift pales in comparison to the upper bound which is over tens of orders of magnitudes higher. Overall, this strengthens our claim that the lower bound is much more faithful to the effective Lipschitz value and can hence serve better to explore various phenomenon observed in over-parameterized neural networks. Lastly, similar distribution plots for other models and datasets can be found in Appendix S3.15.

### S3.8 LIPSCHITZ EVOLUTION FOR OTHER SETTINGS

In this section, we present evolution plots for settings that were mentioned in Section 3.1. We start with evolution plots for FCN networks with MSE loss and then continue with CNN evolution on CIFAR-10 and MNIST. Table S8 shows the values of the corresponding slopes with linear regression $R^2$ metrics for each setup.

Table S8: Estimates of the slopes and linear regression $R^2$ metrics for the upper, **lower** and average norm Lipschitz bounds for various evolution plots.

| Evolution plot | Slopes | $R^2$ metric |
|---|---|---|
| FCN, MNIST1D, CE loss (Fig. 1) | 0.59, **0.46**, 0.44 | 0.9955, **0.9986**, 0.9980 |
| FCN, MNIST1D, MSE loss (Fig. S9) | 0.80, **0.62**, 0.52 | 0.9982, **0.9914**, 0.9991 |
| CNN, CIFAR-10, CE loss (Fig. S10) | 0.22, **0.29**, 0.29 | 0.9461, **0.9327**, 0.9460 |
| ResNet50, subset of ImageNet, CE loss (Fig. 2) | 7.32, **0.49**, 0.48 | 0.8441, **0.9718**, 0.8774 |

From Table S8, one can clearly see how the slope for the upper bound is larger for some more complex models and datasets. Consequently, a simple upper bound is an excessively loose estimator of the Lipschitz constant — despite its theoretical full input domain coverage, this estimator has little practical value due to excessive overestimation. As discussed in detail in Section 3.1, we therefore suggest paying close attention to the lower bound estimator. It would intriguing to see how tighter upper bound estimates compare to our lower and upper bounds, but we leave this investigation for future work.

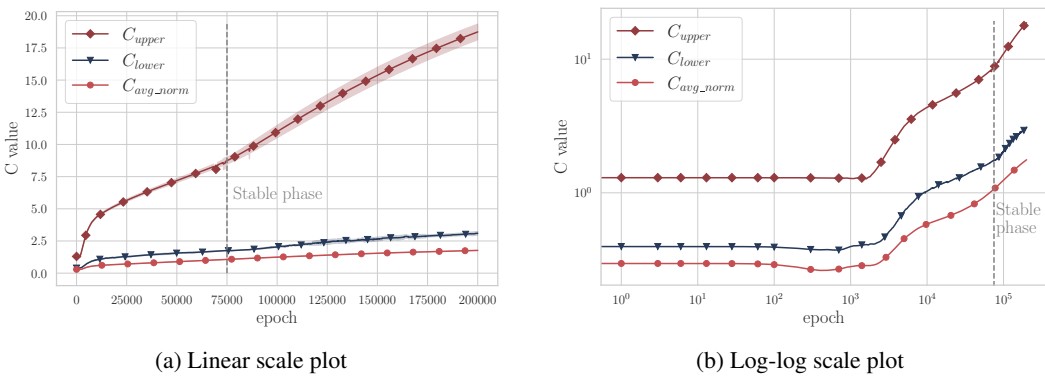

(a) Linear scale plot
(b) Log-log scale plot

Figure S9: Lipschitz constant bounds evolution for **FCN ReLU 256**. The model was trained on **MNIST1D** using **MSE loss** and SGD optimiser. Results are averaged over 4 runs. Stable phase is considered to start after epoch 75,000. We took this model form the Double Descent experiment (see Appendix S2.6.3).

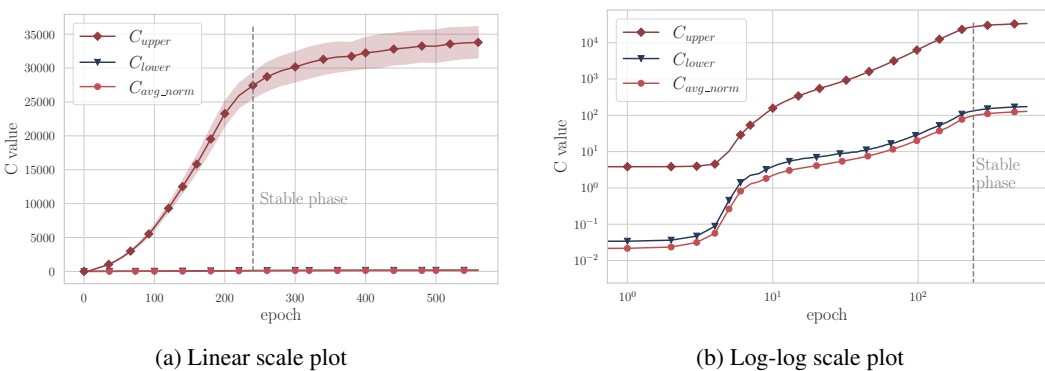

(a) Linear scale plot
(b) Log-log scale plot

Figure S10: Lipschitz constant bounds evolution for **CNN 20**. The model was trained on **CIFAR-10** using Cross-Entropy loss and SGD optimiser. Results are averaged over 4 runs. Stable phase is considered to start after epoch 240. We used the same setup as in the Double Descent experiment (see Appendix S2.6.4).

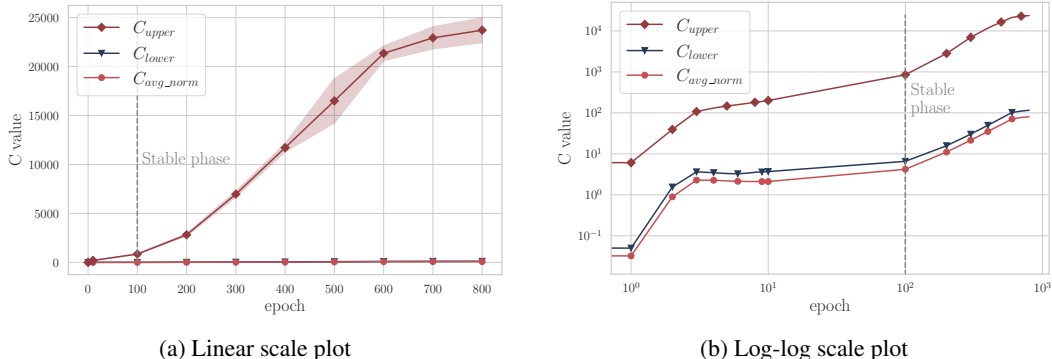

(a) Linear scale plot

(b) Log-log scale plot

Figure S11: Lipschitz constant bounds evolution for **CNN 20**. The model was trained on **MNIST with 10% labels shuffled** using Cross-Entropy loss and SGD optimiser. Results are averaged over 4 runs. Stable phase is considered to start after epoch 100. We took this model form the Double Descent experiment (see Appendix S2.6.6).

## S3.9 LIPSCHITZ DOUBLE DESCENT FOR OTHER SETTINGS

This section showcases Lipschitz constant (left) and train / test loss (right) Double Descent plots for more model class and dataset settings.

*Remark on Figure S12:* Like the test loss, the upper Lipschitz for FCN ReLU networks seems to continue to increase after the second descent, which could potentially be tied to the Triple Descent phenomenon (Adlam & Pennington, 2020), where the test loss shows another peak for $p \approx n^2$. We leave this interesting observation for future research.

*Remark on Figure S16:* For the case of ViT models on CIFAR-10, we also plot the lower Lipschitz constant estimates for a set of 1,000,000 CIFAR-10 convex combinations (denoted as $S^{**}$) to further show the fidelity of the lower Lipschitz bound, even in a Double Descent setting.

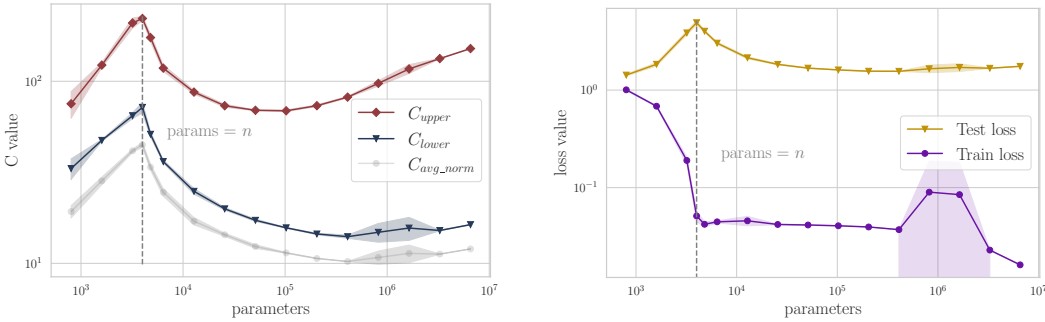

Figure S12: Comparison of various Lipschitz constant bounds with train and test losses with increasing hidden layer width, in the case of **FCN ReLU networks** on **MNIST1D**. Results are averaged over 4 runs. More details about the networks and the training strategy are listed in Appendix S2.6.2.

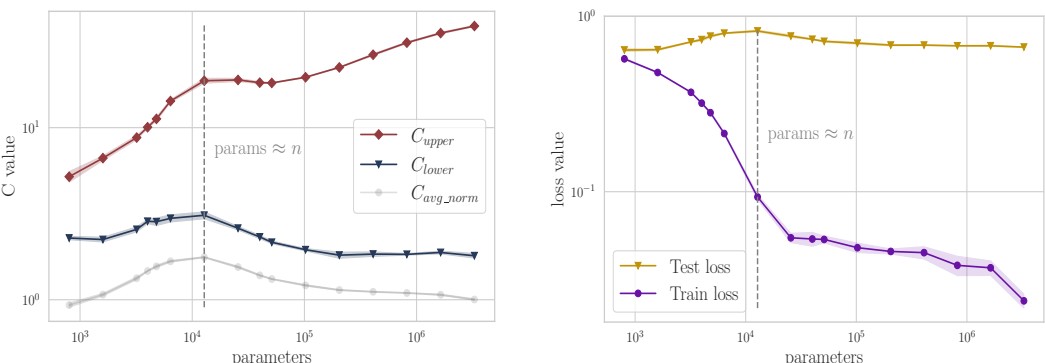

Figure S13: Comparison of various Lipschitz constant bounds with train and test losses with increasing hidden layer width, in the case of **FCN ReLU networks** on **MNIST1D** with **MSE loss**. Results are averaged over 4 runs. More details about the networks and the training strategy are listed in Appendix S2.6.3.

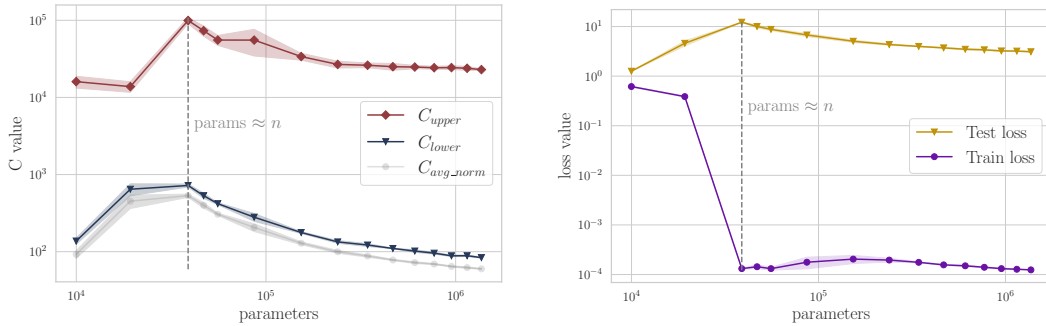

Figure S14: Comparison of various Lipschitz constant bounds with train and test losses with increasing hidden layer width, in the case of **CNN networks** on **CIFAR-10**. Results are averaged over 4 runs. More details about the networks and the training strategy are listed in Appendix S2.6.4.

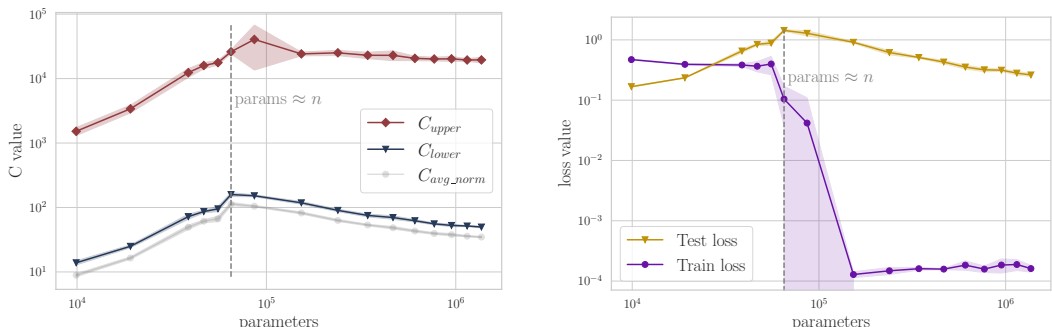

Figure S15: Comparison of various Lipschitz constant bounds with train and test losses with increasing model width, in the case of **CNN networks** on **MNIST with 10% of labels shuffled** with Cross-Entropy loss. Results are averaged over 4 runs. More details about the networks and the training strategy are listed in Appendix S2.6.6.

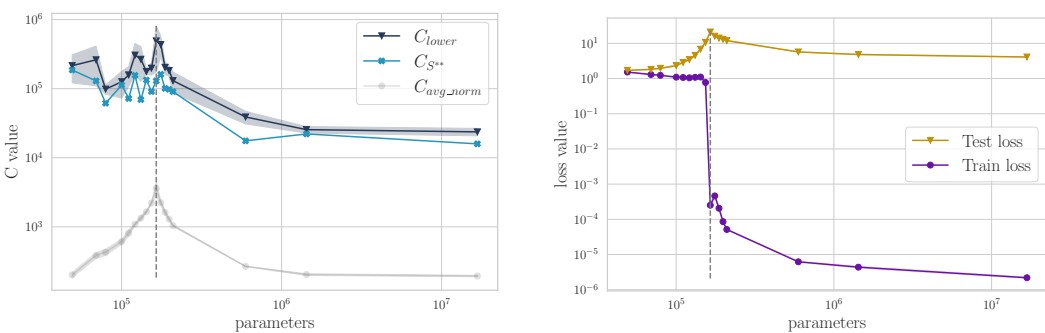

Figure S16: Comparison of various Lipschitz constant bounds with train and test losses with increasing parameter count, in the case of **Vision Transformer networks** on **CIFAR-10** with Cross-Entropy loss. Results are averaged over 4 runs. More details about the networks and the training strategy are listed in Appendix S2.6.7.

## S3.10   MORE EXPERIMENTS ON THE LIPSCHITZ CONSTANT IN THE DOUBLE DESCENT SETTING

This section includes experiments, where we compute Lipschitz constant bounds for a set of models from another study (Singh et al., 2022). Results are produced from training a series of fully connected networks on CIFAR-10 and MNIST with MSE loss. This solidifies our findings on Lipschitz's Double Descent trends.

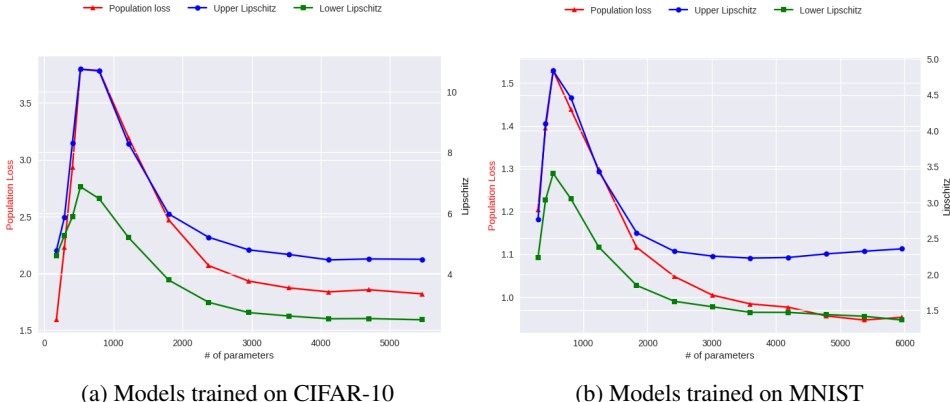

(a) Models trained on CIFAR-10        (b) Models trained on MNIST

Figure S17: Plot of Lipschitz constant bounds for fully connected networks with 1 hidden layer, trained using SGD with MSE loss.

## S3.11   BIAS-VARIANCE TRADE-OFF EVALUATION

In Section 4, we showed the equation for the bias-variance trade-off for the expected test loss. We then proceeded with upper bounding the variance, ignoring the effect of bias. In Figure S18 we display the results of the empirical bias-variance decomposition of the test loss for the MNIST1D Double Descent scenario (see Appendix S2.6.3 for more details). The plot reveals that the Double Descent shape is mostly governed by the variance term, whereas the bias is almost constant across widths.

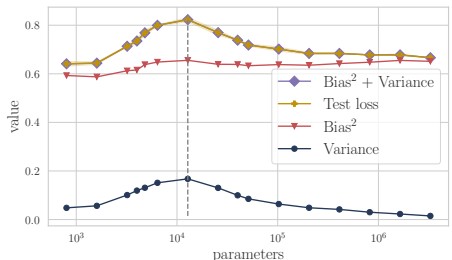

Figure S18: Bias-Variance decomposition of the expected test loss.

## S3.12   DOUBLE DESCENT FOR LIPSCHITZ CONSTRAINED NETWORKS

In this experiment, inspired by the work of Gouk et al. (2021), we replicate the Double Descent scenario with FCN ReLU networks on MNIST1D, where we regularise the product of spectral norms of the weight matrices to be at most $C_{\text{upper}}^{\text{constr}}$. We do this by renormalising each individual weight matrix spectral norm to be at most $\sqrt[\text{depth}]{C_{\text{upper}}^{\text{constr}}}$. In order to preserve the expressivity of highly constrained networks, we multiply network output logits by a temperature $\tau$, as described by Béthune et al. (2022). For $C_{\text{upper}}^{\text{constr}} = 10$, we use $\tau = 10$, for $C_{\text{upper}}^{\text{constr}} = 1$, we use $\tau = 40$, and for no constraint $\tau = 1$ (i.e. no temperature).

To optimise the models, we used SGD with batch size 512 and LR 0.001, with a LR scheduler Cont100. The results are shown in Figure S19. As expected, the test loss follows the Lipschitz constant bounds.

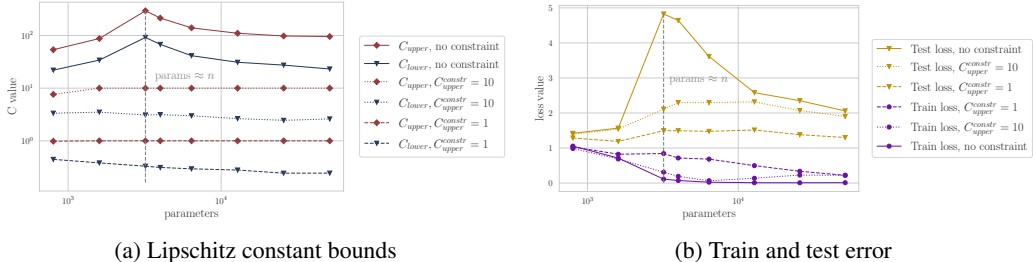

(a) Lipschitz constant bounds        (b) Train and test error

Figure S19: Comparison of various Lipschitz constant bounds with train and test loss with increasing hidden layer width, in the case of Lipschitz-constrained FCN ReLU networks on MNIST1D with different $C_{\text{upper}}^{\text{constr}}$ constraint.

## S3.13 OPTIMISING THE UPPER BOUND FOR THE VARIANCE

As discussed in Section 4 (and more comprehensively in Appendix S4.2), the variance of the learned function is related to the Lipschitz constant by following equation:

$$\mathbb{E}_{\mathbf{x} \sim S'} \text{Var}_\zeta(f_{\boldsymbol{\theta}}(\mathbf{x}, \zeta)) \le 3\,(\overline{C}^2 + \overline{C_\zeta}^2)\,\mathbb{E}_{\mathbf{x} \sim S'}\|\mathbf{x}\|^2 \;+\; 3\,\text{Var}_\zeta(f_{\boldsymbol{\theta}}(\mathbf{0}, \zeta)) \qquad \text{(Var bound 1)}$$

According to Figure 8, variance bounds are rather loose compared to the variance itself. We contribute this difference to the large value of dataset radius $\mathbb{E}_{\mathbf{x} \sim S'}\|\mathbf{x}\|^2$ which can potentially be optimised by choosing a better $\mathbf{x}'$ in bound (Var bound 0). For this case, the optimal minimum for $\mathbb{E}_{\mathbf{x} \sim S'}\|\mathbf{x} - \mathbf{x}'\|^2$ is $\mathbf{x}' = \mathbb{E}_{\mathbf{x} \sim S'}[\mathbf{x}] = \overline{\mathbf{x}}$, which we utilised to recompute the bound in Figure S20.

In the case of MNIST1D, the bound is only almost negligibly better due to $\overline{\mathbf{x}}$ being close to a zero vector for this particular dataset. We leave further analysis of improving the upper variance bound tightness for future work.

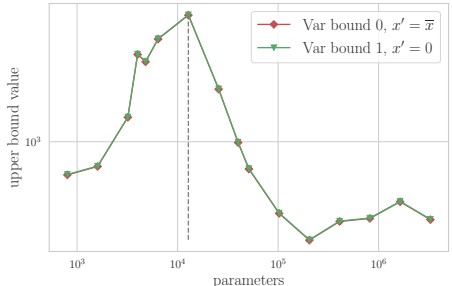

Figure S20: Optimised variance bound compared with (Var bound 1) and variance.

## S3.14 RESNET50 UPPER BOUND ON MORE CHECKPOINTS

Since computing the Lipschitz upper bound is significantly less expensive than the lower bound for the case of ResNet50 evolution, we have additionally evaluated the upper bound on a finer set of checkpoints for one seed to present a more complete picture of the upper bound evolution. According to the results in Figure S21, the upper bound slope after epoch 60 is 7.04 with $R^2$ 0.9624, supporting the representativeness of our previous estimation.

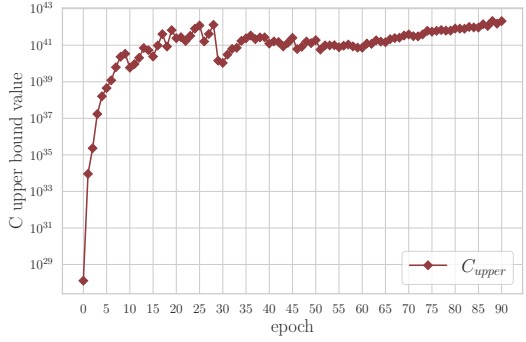

Figure S21: Upper Lipschitz constant bounds evolution for ResNet50. More details in Appendix S2.6.9.

### S3.15 JACOBIAN NORM DISTRIBUTIONS FOR OTHER MODELS AND DATASETS.

In Section 3.2 we have shown a distribution of the norm of the per-sample Jacobian for pretrained ResNet18. Here we provide additional plots for other types of models and datasets. In particular, we showcase the distribution plots for FCNs trained on MNIST1D with Cross-Entropy and MSE losses, CNNs trained on CIFAR-10 and MNIST with 10% label noise and, finally, ResNets on CIFAR-10 (we used pretrained models from Idelbayev). All norms are evaluated on the datasets used for training. Each dotted line represents the maximum norm, or, in other words, the lower Lipschitz bound estimate.

In comparison to just the lower Lipschitz bound estimate, the Jacobian norm distribution plot provides more information on the continuity of the function in the training domain. If the distribution has a long right tail, like in the case of ResNets in Figure S26, the corresponding function has to be rather smooth in the vicinity of most samples, while being more vulnerable to drastic changes for a smaller subset of inputs. It would be fascinating to see if the skewness of this distribution bears a connection to the difficulty of generating adversarial examples, but we leave this direction for future research.

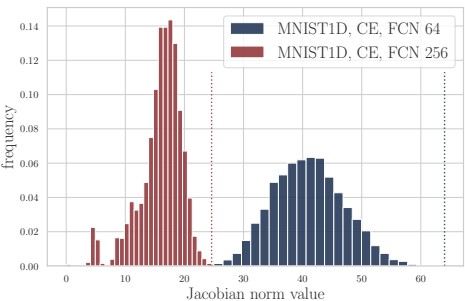

Figure S22: Distribution of the norm of the per-sample Jacobian for FCN ReLU 64 and FCN ReLU 256, trained on MNIST1D with **Cross-Entropy loss**. See Appendix S2.6.2 for training details.

Figure S23: Distribution of the norm of the per-sample Jacobian for FCN ReLU 64 and FCN ReLU 256, trained on MNIST1D with **MSE loss**. See Appendix S2.6.3 for training details.

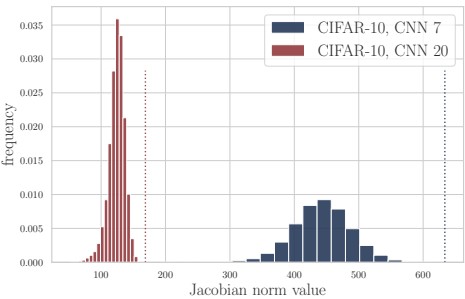

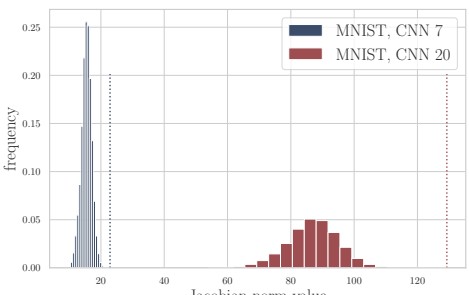

Figure S24: Distribution of the norm of the per-sample Jacobian for CNN 7 and CNN 20, trained on **CIFAR-10**. See Appendix S2.6.4 for training details.

Figure S25: Distribution of the norm of the per-sample Jacobian for CNN 7 and CNN 20, trained on **MNIST with 10% label noise**. See Appendix S2.6.6 for training details.

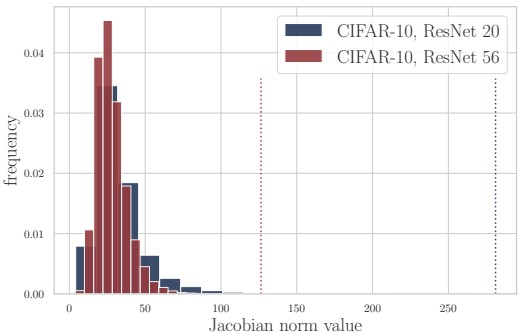

Figure S26: Distribution of the norm of the per-sample Jacobian for ResNet 20 and ResNet 56, trained on CIFAR-10. We refer to Idelbayev for training details.

## S3.16 LOWER LIPSCHITZ BOUND AS A REGULARISER

Although it is arguably easier to use the upper Lipschitz bound as a regulariser due to its lower computational complexity, it could not be the best choice for larger models, where compensating for the exponential increase in the upper Lipschitz estimate by the $\lambda$ hyperparameter might be tricky. In Figure S27, we show the results of training an FCN ReLU network on MNIST1D, while adding $\lambda \cdot C_{\text{lower}}$ a regularisation term. The $C_{\text{lower}}$ estimate is recomputed every epoch using the complete training set.

Although the computational efficiency of this approach leaves much to be desired, we can still see a positive effect on the test loss, *despite almost no change in the upper Lipschitz bounds*.

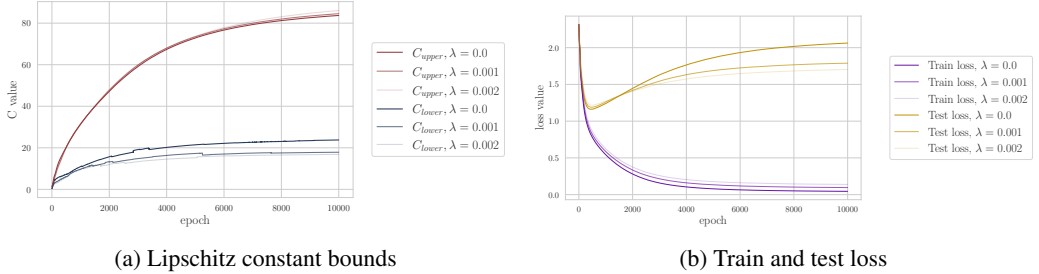

(a) Lipschitz constant bounds

(b) Train and test loss

Figure S27: Plot of Lipschitz constant bounds and train/test loss by training epoch for FCN ReLU network with width 256 trained on MNIST1D. Networks were regularised using various values of the $\lambda$ hyperparameter.

## S3.17 EFFECT OF THE LOSS FUNCTION: CE VS MSE

Surprisingly, training networks with MSE loss results in marginally lower Lipschitz bounds than in the case of Cross-Entropy, as shown in Figure S28a. We contribute this behaviour to the constraints that MSE imposes on the function output in the case of classification. Since in the MSE scenario an ideal model should output a zero vector with only one entry of value 1, model's outputs are restricted to unit vectors for the domain of training samples. At the same time, Cross-Entropy loss does not impose this constraint as output logits are implicitly Softmax-ed. Figure S28b shows that applying Softmax to the output of the Cross-Entropy network shrinks the Lipschitz constant dramatically.

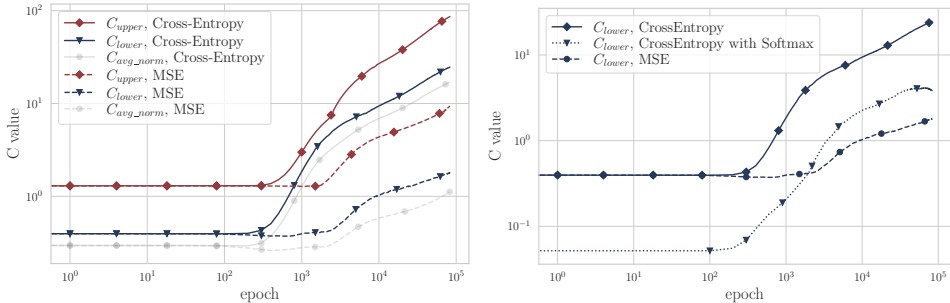

(a) Lipschitz bounds comparison: Cross-Entropy vs MSE

(b) Lower Lipschitz bounds for CE (with and without Softmax) and MSE

Figure S28: Lipschitz constant bounds evolution for FCN ReLU network with 1 hidden layer with 256 neurons, trained using **Cross-Entropy and MSE**. Both models were trained on MNIST1D with SGD, using the same learning rate and LR scheduler. Results are averaged over 4 runs. More details are in Appendix S2.6.12.

### S3.18 EFFECT OF THE OPTIMISATION ALGORITHM: SGD VS ADAM

When a network is trained using the Adam optimiser, Lipschitz constant bounds escalate dramatically compared to the results from SGD. This finding supports the fact that Adam finds solutions that generalise significantly worse, despite great training performance (Wilson et al., 2017; Wang et al., 2021). Note that this trend remains even if we account for Adam's faster convergence, i.e. compare networks at their respective last epochs of training, which are not the same due to various convergence rates (see Figure S29).

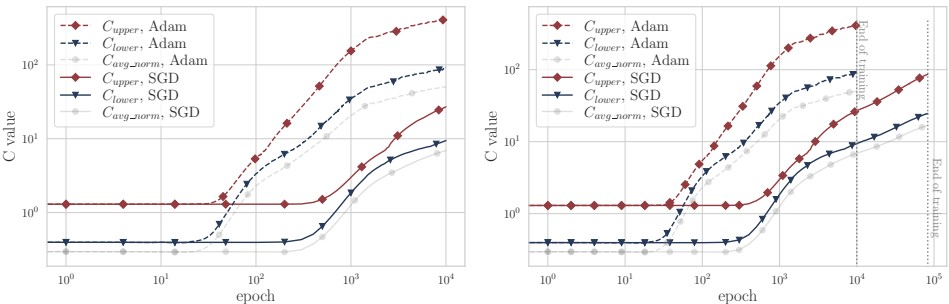

(a) Lipschitz bounds evolution for SGD and Adam **at the same epoch**

(b) Lipschitz bounds evolution for SGD and Adam **at the end of training**

Figure S29: Lipschitz constant bounds evolution for FCN ReLU network with 1 hidden layer with 256 neurons, trained using **SGD and Adam**. Both models were trained on MNIST1D with Cross-Entropy loss, using the same LR and LR scheduler. Results are averaged over 4 runs. We end training when the gradient norm reaches the critical value of 0.01. More details are in Appendix S2.6.13.

We suggest that this behaviour can be explained by observing how far models travel from their initial parameters. Figure S30 shows the evolution of parameter distances (i.e. $param\_dist_\tau = \sum_{t=1}^{\tau} \|\boldsymbol{\theta}^t - \boldsymbol{\theta}^{t-1}\|_2$, where $t$ iterates through saved model checkpoints), which has a similar trend to Lipschitz bounds. In fact, we show that Lipschitz constant can be indeed expressed in terms of parameter distance in our theoretical analysis in Section S4.1. We leave a thorough exploration of this facet for future work. As a bonus we also show that parameter distance also exhibits Double Descent, see Figure S31.

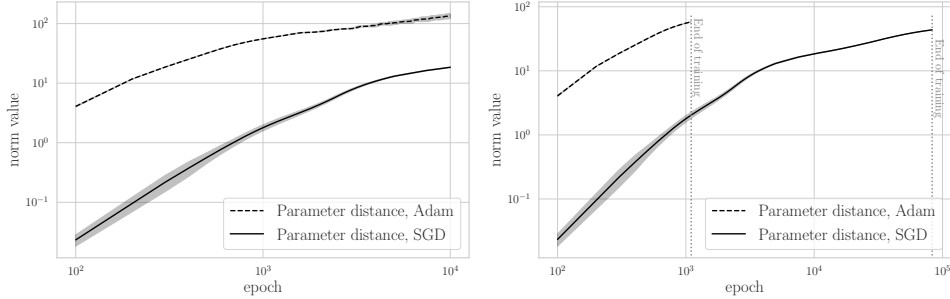

(a) Parameter displacement evolution for SGD and Adam **at the same epoch**

(b) Parameter displacement evolution for SGD and Adam **at the end of training**

Figure S30: Parameter displacement evolution for FCN ReLU network with 1 hidden layer with 256 neurons, trained using **SGD and Adam**. Both models were trained on MNIST1D with Cross-Entropy loss, using the same LR and LR scheduler. Results are averaged over 4 runs. We end training when the gradient norm reaches the critical value of 0.01. More details are in Appendix S2.6.13.

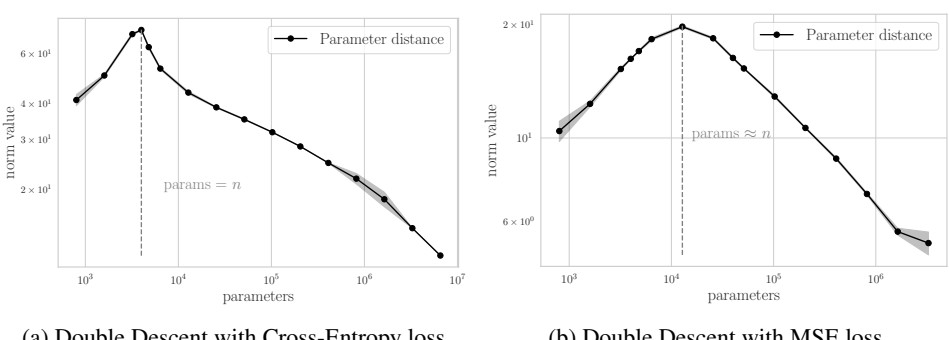

(a) Double Descent with Cross-Entropy loss

(b) Double Descent with MSE loss

Figure S31: Parameter distance at the last epoch for models from the Double Descent on MNIST1D setting with Cross-Entropy (Fig. S12) and MSE (Fig. S13) losses. More details are in Appendices S2.6.2 and S2.6.3.

## S3.19  EFFECT OF DEPTH

To study how depth affects the Lipschitz constant of the network we trained 5 fully-connected networks on MNIST1D with the same learning parameters. According to Figure S32a, all Lipschitz bounds for trained models start to increase with each subsequent layer after 2 layers, following a trend close to power law — $R^2$ linear regression metrics are 0.9749, 0.9483 and 0.8798 for upper, lower, and average Lipschitz bounds respectively. The slopes for the corresponding Lipschitz bounds are 3.33, 2.03 and 1.19, indicating a superlinear trend.

An interesting fact is that the aforementioned trend for lower Lipschitz bounds does not hold for networks at initialisation, both for the case of increasing depth (Figure S32b) and increasing width (Figure S34). Consequently, we see how the effect of feature learning gets manifested in the bounds of the Lipschitz constant and that looking solely at initialisation (as in the style of the lazy regime (Chizat et al., 2019)) would be insufficient. To visualise this 'trend flipping' behaviour we also present evolution plots for the upper and lower Lipschitz constant bounds in Figure S33.

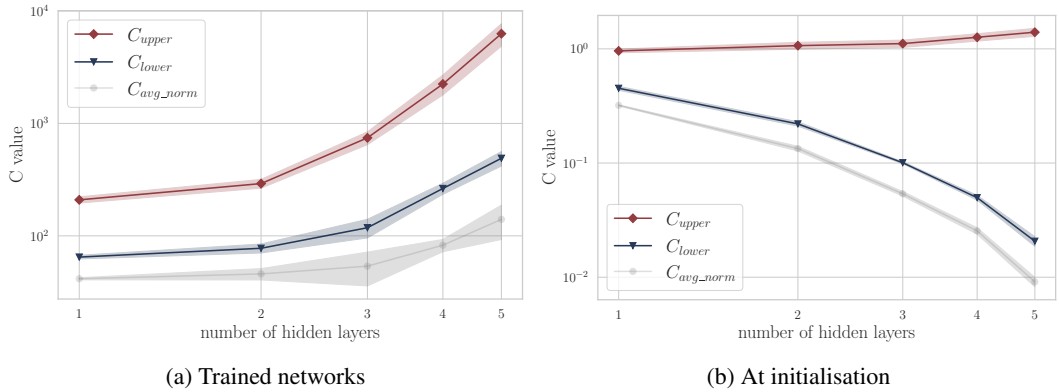

(a) Trained networks

(b) At initialisation

Figure S32: Lipschitz constant bounds for FCN ReLU network with various number of hidden layer with 64 neurons for parameters at initialisation and after training. Results are averaged over 4 runs. More details are in Appendix S2.6.14.

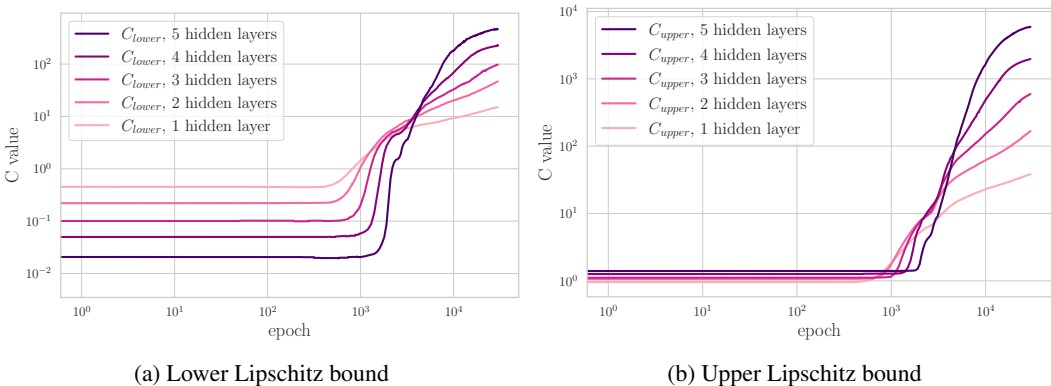

(a) Lower Lipschitz bound

(b) Upper Lipschitz bound

Figure S33: Lipschitz constant bounds evolution for FCN ReLU networks with various number of hidden layer with 64 neurons for parameters. Results are averaged over 4 runs. More details are in Appendix S2.6.14.

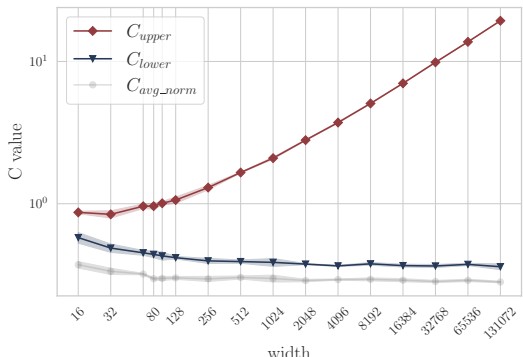

Figure S34: Lipschitz constant bounds for FCN ReLU network with increasing hidden layer width at initialisation. Results are averaged over 4 runs. Here we used models from the Double Descent experiment, see details in Appendix S2.6.2.

## S3.20 EFFECT OF THE NUMBER OF TRAINING SAMPLES

Increasing the number of samples in the dataset results in a corresponding sublinear increase in all Lipschitz bounds. Figure S35 shows the Lipschitz bounds for FCN ReLU 256 networks trained on various random subsets of MNIST1D. Using linear regression we estimated the slope of upper, lower and average Lipschitz bounds to be 0.53, 0.37 and 0.36 respectively, and the $R^2$ metrics are 0.9680, 0.9932 and 0.9928, implying a strong sublinear trend in all Lipschitz bounds. These results suggest that as the number of samples increases, the complexity of the function rises to fit a larger set of points. It would be interesting to precisely tease out this behaviour in terms of relevant theoretical quantities, but we leave that for future work.

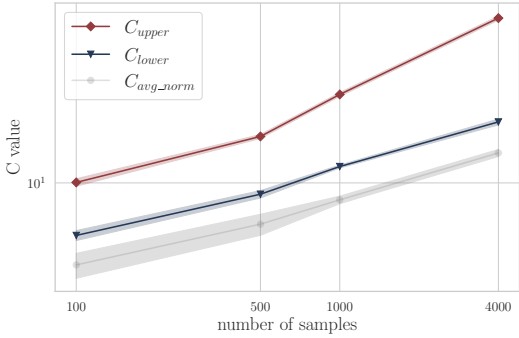

Figure S35: Lipschitz constant bounds for FCN ReLU network with 1 hidden layer with 256 neurons trained for various subsets of MNIST1D. Results are averaged over 4 runs. More details are in Appendix S2.6.15.

## S3.21 EFFECT OF DROPOUT

Intuitively, increasing the probability value in dropout further regularises the model, which should result in lower Lipschitz bounds. To that this hold even for the $C_{\text{lower}}$ metric, we conduct an experiment with a ResNet20 model, trained on CIFAR-10. We impute Dropout layers before each ResNet block (i.e. a group of Convolution and BatchNorm layers that has a skip connection) and before the last Linear layer. The details of the experiment are described in Appendix S2.6.17. Figure S36 shows the results, which indicate that our lower bound metric indeed follows the expected decreasing trend with increasing dropout regularisation.

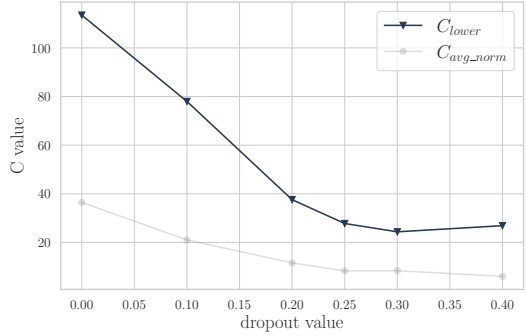

Figure S36: Lipschitz constant bounds for ResNet20, trained on CIFAR-10 with various levels of dropout. More details are in Appendix S2.6.17.

## S3.22 EFFECT OF WEIGHT DECAY

Similar to the previous experiment, we also test the $C_{\text{lower}}$ metric against increasing regularisation via weight decay. Once again, we expect the Lipschitz constant to decrease with stronger regularisation.

The results in Figure S38 are in line with our expectations: the lower Lipschitz bound indeed decreases (with some small noise) with higher values of weight decay.

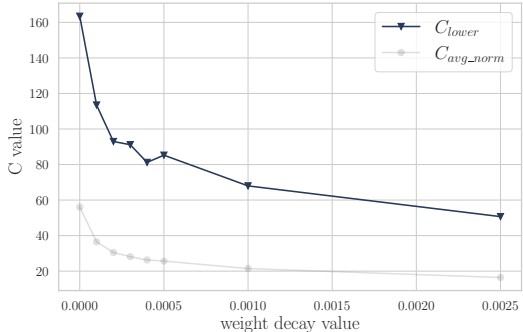

Figure S37: Lipschitz constant bounds for ResNet20, trained on CIFAR-10 with various levels of weight decay. More details are in Appendix S2.6.17.

### S3.23 LOWER LIPSCHITZ BOUND SAMPLES WITH CORRECT AND INCORRECT PREDICTED LABELS

In this experiment, we compute the lower Lipschitz bounds on misclassified and correctly classified labels separately for the case of ResNet20 on CIFAR-10 with 0 dropout and 1e-4 weight decay. As the results demonstrate, incorrectly classified samples indeed more frequently have higher values of the Jacobian norm, compared to the points with correct classes. However, since the lower bound is computed as the supremum, basing the calculation on either of the sets alone would yield a similar value. Moreover, there is simply no correlation between the loss and the Jacobian norm (the value is 0.03), which indicates that there is no clear benefit in only considering misclassified samples for lower Lipschitz bound estimation.

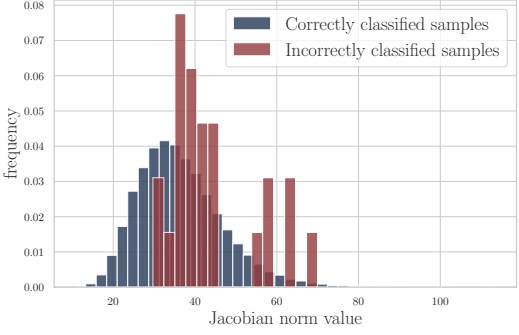

Figure S38: Jacobian norm bounds for points with correct and incorrect labels, predicted by a ResNet20, trained on CIFAR-10 with zero dropout and 1e-4 weight decay. More details are in Appendix S2.6.17.

## S4 THEORETICAL PROOFS

### S4.1 THEORETICAL ANALYSIS OF LIPSCHITZ EVOLUTION

Let us denote $f(\boldsymbol{\theta}, \mathbf{x}) : \mathbb{R}^p \times \mathbb{R}^d \mapsto \mathbb{R}^K$ as our network function, where $\boldsymbol{\theta}$ is our parameter vector. We also denote $\mathcal{L}(\boldsymbol{\theta}, S)$ as our loss and $S$ as our training set. We are interested in finding a bound for $C$ for the trained model at time step $T$:

$$\forall \mathbf{x}, \mathbf{x}' \in \mathcal{D} : \|f(\boldsymbol{\theta}^T, \mathbf{x}) - f(\boldsymbol{\theta}^T, \mathbf{x}')\| \le C\|\mathbf{x} - \mathbf{x}'\| \le C \underbrace{\sup_{\mathbf{x}, \mathbf{x}' \in \mathcal{D}} \|\mathbf{x} - \mathbf{x}'\|}_{=r}$$

**Initial and Final points based analysis.** Let us introduce a simple upper bound on the LHS by adding and subtracting the network at initialisation:

$$
\begin{aligned}
\|f(\boldsymbol{\theta}^T, \mathbf{x}) - f(\boldsymbol{\theta}^T, \mathbf{x}')\| &= \|f(\boldsymbol{\theta}^T, \mathbf{x}) - f(\boldsymbol{\theta}^0, \mathbf{x}) + f(\boldsymbol{\theta}^0, \mathbf{x}) - f(\boldsymbol{\theta}^0, \mathbf{x}') + f(\boldsymbol{\theta}^0, \mathbf{x}') - f(\boldsymbol{\theta}^T, \mathbf{x}')\| \\
&\le \|f(\boldsymbol{\theta}^T, \mathbf{x}) - f(\boldsymbol{\theta}^0, \mathbf{x})\| + \|f(\boldsymbol{\theta}^0, \mathbf{x}) - f(\boldsymbol{\theta}^0, \mathbf{x}')\| + \|f(\boldsymbol{\theta}^0, \mathbf{x}') - f(\boldsymbol{\theta}^T, \mathbf{x}')\| \\
&\le 2C_{\boldsymbol{\theta}}\|\boldsymbol{\theta}^0 - \boldsymbol{\theta}^T\| + C_{\mathbf{x}}(\boldsymbol{\theta}^0)\|\mathbf{x} - \mathbf{x}'\| \\
&\le 2C_{\boldsymbol{\theta}}\|\boldsymbol{\theta}^0 - \boldsymbol{\theta}^T\| + C_{\mathbf{x}}(\boldsymbol{\theta}^0)r \,,
\end{aligned}
$$

where $C_{\mathbf{x}}(\boldsymbol{\theta}^0)$ is the Lipschitz constant in the input space for the model at initialisation and $C_{\boldsymbol{\theta}}$ is the Lipschitz constant for the network in the parameter space. The latter quantity is unfortunately hard to compute, since it requires to search through the space of both parameters and inputs to find the maximum norm. Moreover, it might not be as tight.

**Intermediate-points based analysis.** We can tackle the above issue by applying the same trick iteratively to get a local Lipschitz constant in the parameter space:

$$
\begin{aligned}
\|f(\boldsymbol{\theta}^T, \mathbf{x}) - f(\boldsymbol{\theta}^T, \mathbf{x}')\| &= \|f(\boldsymbol{\theta}^T, \mathbf{x}) + \sum_{t=0}^{T-1} \left(f(\boldsymbol{\theta}^t, \mathbf{x}) - f(\boldsymbol{\theta}^t, \mathbf{x}) + f(\boldsymbol{\theta}^t, \mathbf{x}') - f(\boldsymbol{\theta}^t, \mathbf{x}')\right) - f(\boldsymbol{\theta}^T, \mathbf{x}')\| \\
&\le \sum_{t=0}^{T-1} \|f(\boldsymbol{\theta}^{t+1}, \mathbf{x}) - f(\boldsymbol{\theta}^t, \mathbf{x})\| + \sum_{t=0}^{T-1} \|f(\boldsymbol{\theta}^t, \mathbf{x}') - f(\boldsymbol{\theta}^{t+1}, \mathbf{x}')\| + \|f(\boldsymbol{\theta}^0, \mathbf{x}) - f(\boldsymbol{\theta}^0, \mathbf{x}')\| \\
&\le 2C_{\boldsymbol{\theta}}^{\text{discrete}} \sum_{t=0}^{T-1} \|\boldsymbol{\theta}^{t+1} - \boldsymbol{\theta}^t\| + C_{\mathbf{x}}(\boldsymbol{\theta}^0)\|\mathbf{x} - \mathbf{x}'\| \\
&\le 2C_{\boldsymbol{\theta}}^{\text{discrete}} \sum_{t=0}^{T-1} \|\boldsymbol{\theta}^{t+1} - \boldsymbol{\theta}^t\| + C_{\mathbf{x}}(\boldsymbol{\theta}^0)r \,,
\end{aligned}
$$

where $C_{\boldsymbol{\theta}}^{\text{discrete}} := \sup_{\boldsymbol{\theta} \in \{\boldsymbol{\theta}^0, \dots, \boldsymbol{\theta}^T\}} \sup_{\mathbf{x} \in dom(f)} \|\nabla_{\boldsymbol{\theta}} f(\boldsymbol{\theta}, \mathbf{x})\|$, i.e., the supremum of parameter-wise Lipschitz constants for a discrete set of checkpoints. In comparison to $C_{\boldsymbol{\theta}}$, $C_{\boldsymbol{\theta}}^{\text{discrete}}$ is only evaluated for $(\boldsymbol{\theta}^0, \dots, \boldsymbol{\theta}^T)$, reducing the search space in the parameter dimension (thus it is marked as discrete).

We can further simplify the equation by considering the GD update rule: $\boldsymbol{\theta}^{t+1} = \boldsymbol{\theta}^t - \eta_t \nabla_{\boldsymbol{\theta}^t} \mathcal{L}(\boldsymbol{\theta}^t, S)$ and introducing the bounded gradients assumption (i.e. $\|\nabla_{\boldsymbol{\theta}} \mathcal{L}(\boldsymbol{\theta}, \mathbf{x})\| \le B$). Note that this constraint can be easily fulfilled by using gradient clipping. The term $\eta_t$ denotes the learning rate at time step $t$. Let $\eta$ be the maximum learning rate throughout the epochs. Then we have the following:

$$
\begin{aligned}
\|f(\boldsymbol{\theta}^T, \mathbf{x}) - f(\boldsymbol{\theta}^T, \mathbf{x}')\| &\le 2C_{\boldsymbol{\theta}}^{\text{discrete}} \sum_{t=0}^{T-1} \|\boldsymbol{\theta}^{t+1} - \boldsymbol{\theta}^t\| + C_{\mathbf{x}}(\boldsymbol{\theta}^0)r \\
&\le 2C_{\boldsymbol{\theta}}^{\text{discrete}} \sum_{t=0}^{T-1} \eta_t \|\nabla_{\boldsymbol{\theta}^t} \mathcal{L}(\boldsymbol{\theta}^t, S)\| + C_{\mathbf{x}}(\boldsymbol{\theta}^0)r \le 2C_{\boldsymbol{\theta}}^{\text{discrete}} \sum_{t=0}^{T-1} \eta B + C_{\mathbf{x}}(\boldsymbol{\theta}^0)r \\
&= \left(\frac{2}{r} C_{\boldsymbol{\theta}}^{\text{discrete}} B\eta T + C_{\mathbf{x}}(\boldsymbol{\theta}^0)\right) r
\end{aligned}
$$

Therefore the Lipschitz constant grows in proportion to the number of steps:

$$C \propto \left( \frac{2}{r} C_{\boldsymbol{\theta}}^{\text{discrete}} B \eta \right) T$$

## S4.2    BIAS-VARIANCE TRADEOFF

Let us denote the neural network function as $f_{\boldsymbol{\theta}}(\mathbf{x}, S, \zeta)$ and the ground-truth function as $\mathbf{y}^\star(\mathbf{x})$. Here, $S$ denotes the training set and $\zeta$ indicates the noise in the function due to the choice of random initialisation and the noise introduced by a stochastic optimiser, like stochastic gradient descent (SGD). In other words, one can take $\zeta$ as denoting the random seed used in practice. Then let us assume we have the square loss, i.e., $\ell(\mathbf{x}; f_{\boldsymbol{\theta}}) = \|\mathbf{y}^*(\mathbf{x}) - f_{\boldsymbol{\theta}}(\mathbf{x}, S, \zeta)\|^2$. We can write the loss evaluated on a test set, $S'$, i.e., the test loss, as follows:

$$\mathcal{L}(\boldsymbol{\theta}, S', \zeta) = \mathbb{E}_{\mathbf{x} \sim S'} \left[ \|\mathbf{y}^*(\mathbf{x}) - f_{\boldsymbol{\theta}}(\mathbf{x}, S, \zeta)\|^2 \right] \tag{6}$$

In practice, we typically average the test loss over several random seeds, hence inherently involving an expectation over the noise $\zeta$. We derive a bias-variance tradeoff (Geman et al., 1992; Neal et al., 2018) that rests upon this as the noise source. Also, we consider the fixed-design variant of the bias-variance tradeoff and as a result, we will not average over the choice of the training set sampled from the distribution. In any case, for a suitably large training set size, this is expected not to introduce a lot of fluctuations and in particular, for the phenomenon at hand, i.e. Double Descent, the training set is generally considered to be fixed. Hereafter, for convenience, we will suppress the dependence of the network function on the training set.

Now we do the usual trick of adding and subtracting the expected neural network function over the noise source. Hence, we can rewrite the above as:

$$\begin{aligned}
\mathcal{L}(\boldsymbol{\theta}, S', \zeta) &= \mathbb{E}_{\mathbf{x} \sim S'} \left[ \|\mathbf{y}^*(\mathbf{x}) - \mathbb{E}_\zeta[f_{\boldsymbol{\theta}}(\mathbf{x}, \zeta)] + \mathbb{E}_\zeta[f_{\boldsymbol{\theta}}(\mathbf{x}, \zeta)] - f_{\boldsymbol{\theta}}(\mathbf{x}, \zeta)\|^2 \right] \\
&= \mathbb{E}_{\mathbf{x} \sim S'} \left[ \|\mathbf{y}^*(\mathbf{x}) - \mathbb{E}_\zeta[f_{\boldsymbol{\theta}}(\mathbf{x}, \zeta)]\|^2 \right] + \mathbb{E}_{\mathbf{x} \sim S'} \left[ \|\mathbb{E}_\zeta[f_{\boldsymbol{\theta}}(\mathbf{x}, \zeta)] - f_{\boldsymbol{\theta}}(\mathbf{x}, \zeta)\|^2 \right] \\
&\quad + 2 \mathbb{E}_{\mathbf{x} \sim S'} \left[ (\mathbf{y}^*(\mathbf{x}) - \mathbb{E}_\zeta[f_{\boldsymbol{\theta}}(\mathbf{x}, \zeta)])^\top \left( \mathbb{E}_\zeta[f_{\boldsymbol{\theta}}(\mathbf{x}, \zeta)] - f_{\boldsymbol{\theta}}(\mathbf{x}, \zeta) \right) \right]
\end{aligned}$$

Next, we take the expectation of the above test loss with respect to the noise source $\zeta$ — mirroring the empirical practice of reporting results averaged over multiple seeds. It is easy to see that when taking the expectation, the cross-term vanishes and we are left with the following expression:

$$\mathbb{E}_\zeta \mathcal{L}(\boldsymbol{\theta}, S', \zeta) = \mathbb{E}_{\mathbf{x} \sim S'} \left[ \|\mathbf{y}^*(\mathbf{x}) - \mathbb{E}_\zeta[f_{\boldsymbol{\theta}}(\mathbf{x}, \zeta)]\|^2 \right] + \mathbb{E}_\zeta \mathbb{E}_{\mathbf{x} \sim S'} \left[ \|\mathbb{E}_\zeta[f_{\boldsymbol{\theta}}(\mathbf{x}, \zeta)] - f_{\boldsymbol{\theta}}(\mathbf{x}, \zeta)\|^2 \right] \tag{7}$$

$$= \mathbb{E}_{\mathbf{x} \sim S'} \left[ \|\mathbf{y}^*(\mathbf{x}) - \mathbb{E}_\zeta[f_{\boldsymbol{\theta}}(\mathbf{x}, \zeta)]\|^2 \right] + \mathbb{E}_{\mathbf{x} \sim S'} \mathbb{E}_\zeta \left[ \|\mathbb{E}_\zeta[f_{\boldsymbol{\theta}}(\mathbf{x}, \zeta)] - f_{\boldsymbol{\theta}}(\mathbf{x}, \zeta)\|^2 \right] \tag{8}$$

$$= \mathbb{E}_{\mathbf{x} \sim S'} \left[ \|\mathbf{y}^*(\mathbf{x}) - \mathbb{E}_\zeta[f_{\boldsymbol{\theta}}(\mathbf{x}, \zeta)]\|^2 \right] + \mathbb{E}_{\mathbf{x} \sim S'} \text{Var}_\zeta(f_{\boldsymbol{\theta}}(\mathbf{x}, \zeta)) \tag{9}$$

Overall, this results in the bias-variance trade-off under our setting.

**Upper-bounding the Variance term.**    Now, we want to a do a finer analysis of the variance term by involving the Lipschitz constant of the network function.

$$\text{Var}_\zeta(f_{\boldsymbol{\theta}}(\mathbf{x}, \zeta)) = \mathbb{E}_\zeta \left[ \|\mathbb{E}_\zeta[f_{\boldsymbol{\theta}}(\mathbf{x}, \zeta)] - f_{\boldsymbol{\theta}}(\mathbf{x}, \zeta)\|^2 \right] \tag{10}$$

$$= \mathbb{E}_\zeta \left[ \| \underbrace{\mathbb{E}_\zeta[f_{\boldsymbol{\theta}}(\mathbf{x}, \zeta)] - \mathbb{E}_\zeta[f_{\boldsymbol{\theta}}(\mathbf{x}', \zeta)]}_{a} + \underbrace{\mathbb{E}_\zeta[f_{\boldsymbol{\theta}}(\mathbf{x}', \zeta)] - f_{\boldsymbol{\theta}}(\mathbf{x}', \zeta)}_{b} + \underbrace{f_{\boldsymbol{\theta}}(\mathbf{x}', \zeta) - f_{\boldsymbol{\theta}}(\mathbf{x}, \zeta)}_{c} \|^2 \right] \tag{11}$$

where, we have considered some auxiliary point $\mathbf{x}'$, and added and subtracted some terms. For $n$ vectors, $\mathbf{x}_1, \cdots, \mathbf{x}_n$, we can utilize the simple inequality:

$$\|\mathbf{x}_1 + \cdots + \mathbf{x}_n\|^2 \leq n \sum_{i=1}^{n} \|\mathbf{x}_i\|^2$$

which follows from $n$ applications of the Cauchy-Schwarz inequality. Hence, the variance above can be upper-bounded as:

$$\mathrm{Var}_\zeta(f_{\boldsymbol{\theta}}(\mathbf{x}, \zeta)) \leq 3 \left\| \mathbb{E}_\zeta[f_{\boldsymbol{\theta}}(\mathbf{x}, \zeta)] - \mathbb{E}_\zeta[f_{\boldsymbol{\theta}}(\mathbf{x}', \zeta)] \right\|^2 + 3 \mathbb{E}_\zeta \left\| \mathbb{E}_\zeta[f_{\boldsymbol{\theta}}(\mathbf{x}', \zeta)] - f_{\boldsymbol{\theta}}(\mathbf{x}', \zeta) \right\|^2$$
$$+ 3 \mathbb{E}_\zeta \left\| f_{\boldsymbol{\theta}}(\mathbf{x}', \zeta) - f_{\boldsymbol{\theta}}(\mathbf{x}, \zeta) \right\|^2$$

We can think of $\mathbb{E}_\zeta f_{\boldsymbol{\theta}}(\mathbf{x}, \zeta)$ as the ensembled function mapping, and denote it by saying $\overline{f_{\boldsymbol{\theta}}}(\mathbf{x}) := \mathbb{E}_\zeta f_{\boldsymbol{\theta}}(\mathbf{x}, \zeta)$, and let's assume that it is $\overline{C}$-Lipschitz. On the other hand, let's say that each individual function $f_{\boldsymbol{\theta}}(\mathbf{x}, \zeta)$ has Lipschitz constant $C_\zeta$. Hence we can further reduce the upper bound to

$$\mathrm{Var}_\zeta(f_{\boldsymbol{\theta}}(\mathbf{x}, \zeta)) \leq 3 \overline{C}^2 \|\mathbf{x} - \mathbf{x}'\|^2 + 3 \, \mathrm{Var}_\zeta(f_{\boldsymbol{\theta}}(\mathbf{x}', \zeta)) + 3 \mathbb{E}_\zeta C_\zeta^2 \|\mathbf{x} - \mathbf{x}'\|^2. \tag{12}$$

Now, we bring back the outer expectation with respect to samples from the test set, i.e., $\mathbf{x} \sim S'$:

$$\mathbb{E}_{\mathbf{x} \sim S'} \mathrm{Var}_\zeta(f_{\boldsymbol{\theta}}(\mathbf{x}, \zeta)) \leq 3 \mathbb{E}_{\mathbf{x} \sim S'} \overline{C}^2 \|\mathbf{x} - \mathbf{x}'\|^2 + 3 \mathbb{E}_{\mathbf{x} \sim S'} \, \mathrm{Var}_\zeta(f_{\boldsymbol{\theta}}(\mathbf{x}', \zeta)) + 3 \mathbb{E}_{\mathbf{x} \sim S'} \mathbb{E}_\zeta C_\zeta^2 \|\mathbf{x} - \mathbf{x}'\|^2$$

Notice that while the Lipschitz constant of the neural network function do depend on the training data, the above expectation is with respect to samples from the test set. Hence, we can take the Lipschitz constants that appear above outside of the expectation. Besides, the middle term on the right-hand side has no dependency on the test sample $\mathbf{x} \sim S'$ and so the expectation goes away. Overall, this yields,

$$\mathbb{E}_{\mathbf{x} \sim S'} \mathrm{Var}_\zeta(f_{\boldsymbol{\theta}}(\mathbf{x}, \zeta)) \leq 3 \, (\overline{C}^2 + \overline{C_\zeta}^2) \, \mathbb{E}_{\mathbf{x} \sim S'} \|\mathbf{x} - \mathbf{x}'\|^2 + 3 \, \mathrm{Var}_\zeta(f_{\boldsymbol{\theta}}(\mathbf{x}', \zeta)) \quad \text{(Var bound 0)}$$

where, for simplicity, we have denoted the Lipschitz constant $C_\zeta$ averaged over the random seeds $\zeta$, as $\overline{C_\zeta}$. We can simplify the above upper bounds by taking $\mathbf{x}' = \mathbf{0}$ as the vector of all zeros, resulting in:

$$\mathbb{E}_{\mathbf{x} \sim S'} \mathrm{Var}_\zeta(f_{\boldsymbol{\theta}}(\mathbf{x}, \zeta)) \leq 3 \, (\overline{C}^2 + \overline{C_\zeta}^2) \, \mathbb{E}_{\mathbf{x} \sim S'} \|\mathbf{x}\|^2 + 3 \, \mathrm{Var}_\zeta(f_{\boldsymbol{\theta}}(\mathbf{0}, \zeta)) \quad \text{(Var bound 1)}$$

