# OpenReview forum: "Some Fundamental Aspects about Lipschitz Continuity of Neural Networks"
_ICLR.cc/2024/Conference — ICLR 2024 poster_

### Official Review · Reviewer_Sk7T · 2023-10-25

**Soundness:** 2 fair
**Presentation:** 2 fair
**Contribution:** 1 poor
**Rating:** 3
**Confidence:** 4

**Summary:**

This paper presents some empirical studies on the Lipschitz constant of neural networks. The studies reveal three major results:
1. the evolution of Lipschitz constant during training; 2. the double descent of Lipschitz constant of neural networks; 3. The Lipchitz contant variation with respect to random labels. The study also shows the fidelity of lower Lipschitz bound.

**Strengths:**

The perspective and results are interesting: the Lipschitz constant of a function is an intrinsic property of neural networks, and it has connections with many learning theoretical properties. For example, the double descent of phenomenon of neural networks is known and understanding this phenomenon from the Lipschitz constant evolution can be an interesting persepctive.

**Weaknesses:**

1. Overall this paper lacks coherence. Though this paper studies the Lipschitz constant of neural networks, each question studied in the paper is quite independent. Also these questions are important and each deserves an in-depth study. The empirical result is interesting yet insufficient to understand the phenomenon per se.

2. The paper is also not rigorous. For example, in the intro, the paper stated that "to put it more accurately, it is the maximum absolute change in the function per unit norm change in the input". This statement is only true when the function is scaling invariant. Also because of each question is not studied thourouly, many of the aspects are not rigorously studied. for example, the paper studies effective Lipschitz constant but in reality distributional shift may occur and how the effective Lipschitz constant may change is not known. The scope of this paper is too big and each of the questions requires an in-depth analysis.

3. The paper is also not well-organized. This comes from that the paper lacks coherence. Even though each point is explained, it is still unclear about the message of this paper.

**Questions:**

The layerwise upper bound is known a loose measurement, and indeed there is a giant gap between the upper and lower bounds in the experiments. Is it possible to strengthen the measurement to have a more precise characterization of the Lipschitz constant?

---

> ### Author Response · Authors · 2023-11-21
> **Response**
>
> Thank you for your critical review of our paper. We were glad to see that you have found that our offered **“perspective and results are interesting”**. In the response below we will go through your questions and other concerns, and, hopefully, alleviate all of them.
>
> &nbsp;
>
> ## Coherence
> - On first glance, these questions might seem independent, but to us, having battled through these questions, we think that **the presented questions are the three most important and intriguing aspects that come naturally** in the context of Lipschitz continuity in neural networks.
> - We agree that each of them could deserve their own in-depth discussion, but it should be emphasized that **none of these aspects can be completely divorced from each other**. For instance, it is hard to measure generalization or optimization aspects of Lipschitz, without first affirming the fidelity of its bounds (as we have done here with the lower Lipschitz). Neither can one get a complete picture of the effect of Lipschitz on generalization, without understanding the role of the noise contained in the data distribution.
> - Thus, our work strived to put together these various interdependent aspects of the Lipschitz together, and more generally cover the breadth of its effects. We agree, in no uncertain terms, that studying the finer intricacies of each individual question is crucial --- but one cannot paint the finer strokes unless one first utilizes the broad strokes and brushes to obtain a proper sight and outline of the concerned subject. And we believe, while not all the finer strokes are in there yet, we have done our best to comprehensively outline the subject of Lipschitz continuity in neural networks.
>
>
> &nbsp;
>
> ## paper is also not rigorous
> We do not quite agree with this stance. Please allow us to make our case:
>
> 1. Firstly, in the introduction, we just wanted to appeal to a broader audience and attempted to introduce the concept of Lipschitz continuity in simple terms, without encumbering the reader. But we are happy to add the mentioned caveat.
>
> 2. Other reviewers also hold a similar stance as us. In particular, we would to note the mentions by:
> - Reviewer Jyw9: “conducted extensive experiments to showcase its findings and offers a comprehensive exploration of experimental details and discussions”.
> - Reviewer EeTb: “Extensive experiment to show the different aspects in the discussion”.
> - Reviewer F3fH:  “ it covers the topic pretty well”, “A lot of experiments in various regime with many different real life instances are quite convincing”.
>
> 3. Also, for instance, your very interesting suggestion on Distributional shift has already been studied in **Section S3.2 and S3.3** of the Appendix (in the form of  Adversarial examples and Out-Of-Distribution noise sampling experiments).
>
> While the presence of all the supplementary results and experiments in the Appendix might have led to your inference, we sincerely believe that our work is one of the most comprehensive studies on this topic. We will be more than happy to further streamline the organization of the supplementary results, so that this point gets clarified to the reader.
>
> &nbsp;
>
> ## Method's precision
> We have made several attempts to tackle this, by trying to compute local Lipschitz in several scenarios in a bid to increase its value (such as with convex combinations, or additive Gaussian noise, and even adversarial perturbtations). But these interventions still showed the fidelity of the original lower Lipschitz formulation. In general, this is an important open question.
>
>
> &nbsp;
>
>
> *We hope that the response above comprehensively addresses your concerns. However, if something remains unclear or if there are any further questions, we will gladly answer them. Lastly, in light of this response, we also earnestly hope that you will reconsider your review score.*

---

> ### Comment · Reviewer_Sk7T · 2023-11-22
> **Thanks for the response**
>
> Dear authors,
>
> I appreciate your response. Could you list your research questions (i.e., what your empirical evaluations try to address), what are your empirical findings, and how they address these questions?

---

> ### Author Response · Authors · 2023-11-22
> **Reply: Part 1**
>
> Thanks for your prompt reply. Sure, below we list the key research questions which we tackle in our paper, what are our empirical findings, and how the corresponding empirical findings are a step forward in addressing them.
>
> But, before we delve into that, to put it a single line, *the unifying theme across all these questions is **understanding how the interaction between over-parameterization and the Lipschitz continuity of the function pan out**.*
>
> &nbsp;
>
> ----
>
> &nbsp;
>
> **Question 1.** One of our primary questions was *how to measure the Lipschitz constant to good fidelity while being efficient for modern over-parameterized networks* on large-scale datasets.
>
> We also wanted such a measure to be valid throughout training. (This made us wonder, as a sub-question, about the dynamics of the Lipschitz continuity of neural networks: what is it like at initialization and how much does it develop during training?)
>
>
>   - **Empirical evaluation and findings:** Our experiments in Sections 3.1 and 3.2 probed how well the lower Lipschitz bound, evaluated on a finite set of samples, would form a good proxy of the effective Lipschitz constant. This was motivated by the fact that, theoretically, it would approach the effective Lipschitz constant in the limit of the number of samples drawn from the input distribution. But, remarkably, we observed its reliability to serve as a measure of the effective Lipschitz, since it changed by only a relatively feeble amount in the presence of varying inputs by considering the convex combinations of data points or with additive Gaussian noise.
>
>
>   - **Implications:**
>     - Consequently, this suggested *a practical and reliable way to gain insights into the Lipschitz behaviour* of neural networks, that was valid throughout training.
>     - But more than that, while there might be some measure-zero (or simply, ‘wild’)  sets where the function behaviour could be arbitrarily complex and the Lipschitz hard to ascertain, *effectively on the distribution the Lipschitz constant is determined by ‘tricky’ points lying near the decision boundary*, as shown in Section 3.3
>
> &nbsp;
>
>
> **Question 2.** The natural question, which we could pursue once the above question had been reasonably settled, was *if the Lipschitz constant was a suitable indicator of generalization for modern over-parameterized networks*.
>
> This was motivated by the fact that, in several learning theory papers, one can often notice the occurrence of the Lipschitz constant in the generalization bounds. However, the relevance of these bounds had been put into question by the double descent phenomenon, especially since the nature of the Lipschitz constant was largely obscure previously.
>
>
>   - **Empirical evaluation and findings:** In Section 4, we noticed that both the lower and upper Lipschitz showed a *double descent trend that aligned tightly with the double descent behaviour in the test loss* (see Figure 7) in their non-monotonicity.
>
>   - **Implications:**
>      - Firstly, this *reaffirmed that the Lipschitz behaviour is still relevant to the generalization* abilities of neural networks (as well as reaffirming the traditional learning-theoretic works).
>      - Besides, the observation of double descent at the level of the Lipschitz constant had wider implications than the usual double descent in the test loss since Lipschitz continuity also inherently characterizes the robustness of the network predictions to input perturbations. Hence, this would suggest that *over-parameterized networks would also fare better in the presence of noise (adversarial or o.o.d.) than networks with fewer parameters*, which we also noticed in Sections S3.2 & S3.3.
>
> &nbsp;
>
> *(continued below)*

---

> > ### Author Response · Authors · 2023-11-22
> > **Reply: Part 2**
> >
> > **Question 3.** Subsequently, we wanted to understand *to what extent would the better ‘noise-resilience’ properties of over-parameterized networks hold*.
> >
> > A popular way, given the impactful Zhang et al. (2017, 2021) work,  about over-parameterized networks being perfectly capable of fitting points with random labels, was to analyze this in the context of label noise. In other words, when under/over-parameterized networks are forced to fit noise, whether they do so in a benign manner, i.e., the effect of fitting noise is confined locally and does not affect the overall function, or does fitting the noise results in a malignant change in the kind of function learnt.
> >
> >   - **Empirical evaluation and findings:** In Section 5, our empirical evidence suggested how there is a more subtle interplay of the amount of label noise and over-parametrization here, as there is a double descent like non-monotonic behaviour of the lower Lipschitz as well as the test loss, with the amount of label noise. (see Figure 10)
> >  - **Implications:**
> >     - This implies that heavily over-parameterized $p>>n$ networks fit random labels more smoothly, compared to networks with fewer parameters.
> >     - Mere over-parameterization $p\approx n$ alone does not guarantee fitting random labels perfectly. But, to do so, the amount of over-parameterization needed is dependent upon the amount of label noise present.
> >
> >
> > &nbsp;
> >
> > **Remark.**  If we contrast questions 2 and 3, the former is more of an observational study, while the latter is an interventional study about the relation of the generalization with the Lipschitz behaviour of over-parameterized networks, hence further strengthening our broader findings.
> >
> > &nbsp;
> >
> > -----
> >
> > &nbsp;
> >
> > *References:*
> > - Zhang, Chiyuan, et al. "Understanding deep learning requires rethinking generalization.", ICLR 2017.
> > - Zhang, Chiyuan, et al. "Understanding deep learning (still) requires rethinking generalization.", Communications of the ACM 64.3 (2021): 107-115.

---

### Official Review · Reviewer_EeTb · 2023-10-28

**Soundness:** 3 good
**Presentation:** 3 good
**Contribution:** 3 good
**Rating:** 6
**Confidence:** 2

**Summary:**

The paper discusses Lipschitz continuity of neural networks in the following aspects. Firstly, they show the fidelity of the local Lipschitz-based lower bound, by providing several experiments on different models and datasets (section 3.2, also in appendix), and provide their intuition based on a toy example (section 3.3). With that, they investigate the trend of the Lipschitz constant, from initialization and during the course of training. Secondly, they discuss the (implicit) Lipschitz regularisation and double descent behavior, mainly in the context of over parameterization setup of the deep models and a bias variance trade off argument is provided. Thirdly, they discuss about the Lipschitz constant in the presence of the label noise, with the main focus of the network capacities vs the noise strengths. A hypothesis is provided: “while the network is able to fit the noise, Lipschitz constant should increase”, beyond which the network will reach a memorization threshold, and collapse to a smoother function with a smaller Lipschitz constant. In all the 3 aspects discussed, empirical evidence is provided based on several models and datasets.

**Strengths:**

Extensive experiment to show the different aspects in the discussion, as well as to provide evidence for their intuition and hypothesis.

**Weaknesses:**

Less theoretical explanation of the different aspects discussed.

**Questions:**

1. Is it possible to provide theoretical explanation of the different aspects found?
2. Is it possible to extend some of the experiment in any of the language models?

---

> ### Author Response · Authors · 2023-11-21
> **Response**
>
> Thank you for your review and for acknowledging the **extensiveness of our experiments**. We will address your concerns in the response below.
>
>
> &nbsp;
>
>
> ## Theoretical explanations:
> - The focus of our work, as we have explicitly stated, is primarily experimental, i.e. an elaborate investigation of the behaviour of the Lipschitz constant in a vast set of scenarios covering various optimization, datasets, and architectural choices. In this regard, we should also bear in mind **reviewer F3fH’s** comment that *“[this] is an easy weakness to raise for any experimental paper”*.
> - Nevertheless recognizing the value of theoretical insights, we have attempted to offer, wherever possible, useful *theoretical arguments to complement these insights*, such as those in sections 3.1 under “A theoretical picture” and section 4 “A bias-variance trade-off argument”, whose proofs are developed in more detail in appendix S4.1 and S4.2 respectively.
> - But surely, these are *not yet definitive*. Another concrete idea for an in-depth investigation, that we would have liked to explore, was to utilize the general form of Tikhonov regularization (Bishop 1995), in terms of the function Jacobian, when learning under noise and relate it to the noise spectra characteristics used inherently in various proofs of benign overfitting (Bartlett, et al. 2020).  But, given the current scope, we were forced to leave a thorough theoretical investigation to the next version of the paper.
>
> To conclude, we believe that our empirical findings, together with the seeds for theoretical analyses, will serve as useful scaffolding for other researchers to develop more thorough theoretical explanations and, in general, foster future research, as noted by **reviewer Jyw9**.
>
> Bishop, Chris M. "Training with noise is equivalent to Tikhonov regularization." Neural computation 7.1 (1995): 108-116.
> Bartlett, Peter L., et al. "Benign overfitting in linear regression." Proceedings of the National Academy of Sciences 117.48 (2020): 30063-30070.
>
> ## Extension to Language Models
> - Given their increasing relevance in the scientific community and broader society, experimenting with language models is another exciting venue, but something which we had to sadly leave out for future investigations, due to limited time.
> - Apart from the involved computational challenges in approximating Lipschitz for large language models, which could potentially be carried out without as much difficulty via our lower Lipschitz bounds, this experiment also requires reflecting on and honing upon the right definition of the Lipschitz constant in the language domain. Should it be defined based on token embeddings or should it be based on (semantic) perturbations of textual output?
> - For now, we would like to refer you to our results for the Vision Transformer model in Figures S7 and S16, which potentially suggest the nature of the Lipschitz behaviour in Transformers in the language domain.
>
> All in all, this is an exciting question and we hope to include these results in future iterations of this work.
>
> &nbsp;
>
> *We hope our response helped to clarify your questions. If something remains unclear, we are more than happy to discuss further.*

---

> > ### Comment · Reviewer_EeTb · 2023-11-22
> > **Thanks for response**
> >
> > Appreciate the response. Thank you

---

### Official Review · Reviewer_Jyw9 · 2023-10-29

**Soundness:** 3 good
**Presentation:** 3 good
**Contribution:** 3 good
**Rating:** 6
**Confidence:** 3

**Summary:**

This paper focuses on the importance of Lipschitz continuity in neural network models, which influences their robustness, generalization, and susceptibility to adversarial attacks. In contrast to previous research that aims to tighten Lipschitz bounds and enforce specific properties, this study delves into characterizing the Lipschitz behavior of Neural Networks. It conducts empirical experiments across various scenarios, including different architectures, datasets, label noise levels, and more, exploring the limits of lower and upper Lipschitz bounds.

Notably, the paper highlights the strong adherence to the lower Lipschitz bound, identifies a noteworthy Double Descent trend in both upper and lower bounds for Lipschitz continuity, and offers insights into how label noise affects function smoothness and generalization.

**Strengths:**

- This paper conducted extensive experiments to showcase its findings and offers a comprehensive exploration of experimental details and discussions.
- The paper raised intriguing facets of Lipschitz continuity within neural network models, which are likely to attract substantial interest from the deep learning community aiming to develop theory and practical algorithms based on these observations.

**Weaknesses:**

- While the paper provides thorough experiments and in-depth discussions, its novelty might be subject to question. As also mentioned in the paper, there is concurrent research with a similar focus that has also highlighted the connection between the Lipschitz constant and Double Descent, although they tracked only an estimate of the Lipschitz constant. I appreciate the authors’ efforts in sharing more empirical observations and discussions. But I am not very confident that the paper is completely novel.

- From the optimizer's perspective, this paper may have some shortcomings as it does not delve into the discussion of weight decay and dropout, which are widely employed regularization techniques in neural network training. It is noted in the paper that the intention is not to focus on regularization techniques, but there are a few concerns to address. The term "weight decay" is mentioned only once in S2.6.9 (Page 22) with non-zero values. This implies that all other experiments either employ zero weight decay or the paper lacks sufficient implementation details. Similarly, the term "dropout" appears only once in S2.6.7 (Page), and the paper does not provide any insights or discussions regarding the impact of dropout.

**Questions:**

- As highlighted in the weaknesses section, my primary concern revolves around understanding how commonly-used and seemingly simple regularization techniques like "weight decay" and "dropout" influence the Lipschitz constants. The paper would benefit from thorough discussions in this regard, rather than solely focusing on scenarios without regularization.

- In the context of classification problems, it would be advantageous for the paper to display C_lower values separately for mis-classified samples and correctly-classified samples.

---

> ### Author Response · Authors · 2023-11-21
> **Response**
>
> Thank you for your constructive feedback. We are happy to see that you found the paper had a “comprehensive exploration” and that it **”is likely to attract substantial interest from the deep learning community aiming to develop theory and practical algorithms”**. Here, we will address your concerns one by one.
>
> &nbsp;
>
> ## Novelty
> Our paper indeed bears similarities with some concurrent works (some occurring publically within a span of as few as ~ 20 days), and we have been quite upfront about it. But we believe there are still some notable differences, in regards to these works, due to which our paper provides for a rounder and more complete picture.
>
> 1. By our extensive focus on empirically verifying, at scale, **the fidelity of the lower Lipschitz bound to the effective Lipschitz constant**, as well as the general approach of sandwiching the true Lipschitz, our results provide a direct and almost unequivocal implication about the Lipschitz behaviour of deep neural networks.
> 2.  **This paper unifies many aspects around the Lipschitz constant**, which albeit not as matured as here are scattered through the literature, and provides a holistic treatment with a thorough analysis. We strived to reflect all the key aspects that one would naturally expect the Lipschitz constant to have a bearing on, such as the architectures, datasets, noise levels, optimizers, and possible regularizations (see below).
> 3. More specifically, we present an **expanded view upon concurrently observed concepts** (like the shift of the interpolation threshold in Double Descent in the presence of noise) and put them into a more general framework that gives a more complete picture of the phenomena (i.e. relation with label-noise-wise Double Descent, see Section 5). Wherever possible, we have also tried to complement the empirical findings by sketching theoretical insights and providing useful intuition via synthetic examples.
>
>
> Therefore, we strongly believe that our insights would be valuable in fostering further research on the Lipschitz properties of neural networks, despite concurrent work.
>
>
> &nbsp;
>
>
> ## Optimizer’s perspective …. Fresh experiments on weight decay and dropout:
>
>
> We agree with you, this is a very valid concern. While we briefly discuss the effect of explicit Lipschitz regularization schemes, most of our experiments do not use any weight decay, except for the ResNet or ViT on ImageNet, and neither dropout. Given that these are some of the most frequently employed regularization methods in practice, it would be very interesting to see if they have an implicit effect on the Lipschitz or not. Without further adieu, these results can be found below:
>
>
> **New experiments and results:** We have taken this opportunity to expand our Lipschitz investigation to the cases of increasing weight decay and dropout probability for the case of ResNet20 on CIFAR-10. We include our results in the new version of the paper in Appendices **S3.21 and S3.22**.
>
>
> - The key finding is that as the regularization strength increases (either through weight decay or dropout), the lower Lipschitz bound decreases.
> - Hence, this provides a helpful assurance that even usual regularization strategies have an indirect effect of regularizing the Lipschitz behaviour.
> - While this is to be expected in the case of a linear model, where the Lipschitz constant is just the norm of weights (and enforcing weight decay would be exactly equivalent to a Lipschitz regularization), it is interesting to see that this effect seeps through for networks trained in practice.
>
>
> ## C_lower separately for correctly-classified and misclassified samples.
> That’s a great suggestion. We have gone ahead and carried out this experiment, the results of which can be found in Section S3.23, page 42. We notice that while incorrectly classified tend to more frequently result in higher Jacobian norms, as you were hinting at, there are also correctly classified points with similar levels of high Jacobian norms. As the local Lipschitz definition is based on a supremum, basing the calculation on either of the sets alone would yield a similar value. Although, it seems to offer an empirical advantage, where one can prioritize computation of Jacobian on incorrectly classified when resource bound.
>
> &nbsp;
>
> *We hope that our response helped alleviate the concerns you had about our paper. Should you have any further questions or comments, we remain at your disposal. If we have resolved your concerns, we would be glad if you would consider raising your score.*

---

> ### Comment · Reviewer_Jyw9 · 2023-11-22
>
> Dear authors,
>
> Thank you for your detailed response! It has addressed almost all my concerns. While I intend to increase my score at this moment, I would like to hold on a bit for the upcoming AC/Reviewer discussion phase.
>
> The paper is well-written overall. However, my primary concern is the appropriateness of a 45-page compilation of multiple empirical studies for publication in the ICLR conference proceedings. In particular, as also pointed out by Reviewer F3fH, the paper is mostly located within its own appendix.
>
> I will work closely with other reviewers and AC in the next discussion phase to make a final decision.
>
> Thanks!

---

> > ### Author Response · Authors · 2023-11-22
> > **Thanks and pointer to global response**
> >
> > Thanks for your response, and we are glad to hear that it has addressed almost all your concerns.
> >
> > We would like to refer you to the global response https://openreview.net/forum?id=5jWsW08zUh&noteId=LJR4Q0ihbF, which we have posted in relation to this concern.

---

### Official Review · Reviewer_F3fH · 2023-10-31

**Soundness:** 4 excellent
**Presentation:** 3 good
**Contribution:** 3 good
**Rating:** 8
**Confidence:** 4

**Summary:**

The paper focus on the Lipschitz constant of Neural Networks, and in particular focus on different aspect: 1) how and what do we learn from different Lipschitz bounds from the literature and in particular the gap between lower and upper bounds and its evolution during training, 2) the impact of over-parametrisation on the Lispchitz constant and 3) the impact of noisy labels the network lipschitzness.
The author focus on an intense experimental protocol in order to shade some lights on many claims about the Lipschitz constant of neural networks.

**Strengths:**

The paper is very nice to read (if the appendix had been printed aside), it covers the topic pretty well and its relationship with related works is well described.

It is mostly an experimental study and it seems to me that the methodology is correct. A lot of experiments in various regime with many different real life instances are quite convincing to me. All experiments lead to a reasonable or theoretically supported interpretation.
I quite appreciate that many experiences are real life: with trained network of descent size on well known datasets.

**Weaknesses:**

The article covers many subject and proposes many illustrations of their finding. However a limitation of this work relies in the lack of theoretical insights on the different findings that are discussed (which is an easy weakness to raise for any experimental paper, I admit).

Reading this article is a constant back-and-forth between the main article and its appendix. It often feel that the main article is a glossary to the appendix. As such it often feels like the article should be 'vectorized' and would be better put into a journal format.
For this reason I find difficult to correctly judge the adequacy of this (paper+appendix) to a conference like ICLR and I am open to this discussion with my fellow reviewers and AC.

Bibliography:

- many references are incomplete: they point at arXiv versions rather than their peer-reviewed publications.
- Some references are doubled such as 'On lazy training in differentiable programming' by Chizat et al.
- Are you certain that the reference to (Gomez et al. 2020) ('Lipschitz constant estimation of Neural network via sparse polynomial optimization') shouldn't be (Latorre et al.) as I do not see the name 'Gomez' in the original paper.

**Questions:**

Definition 2.2 is more of a proposition following Definition 2.1.

Section 3.2: How do you compute the convex combinations of the domain samples? Does it actually makes a lot of difference with looking at random points around the samples (say a gaussian centered at a specific sample)? Intuitively it is where one could find the steepest parts of the neural networks rather than convex combinations.

Figure 8: does the practical variance follow the theoretical bound

More generally and in the actual context of the field, it would be very interesting to add experiments with Transformers (with constraint inputs to make it Lipschitz).

**Appendix:**

S4, Intermediate-points based analysis:
I am not sure about how to go from 2nd line to 3d line of the list of inequalities. It seems to me that $C^{discrete}$ is not the correct constant to consider and in order for the inequality to be correct, the supremum should be considered on all $\theta$. It however doesn't seem to be critical to the result about the growth of the Lipschitz constant through training. Could you confirm or dismiss this comment?

S3.17: I am very surprised with this and I agree with the comment made there, do you have any explanation to propose for such a marginal difference?

S3.18: Have you try to apply the same methods as in S4.1 but for Adam (or maybe RMSprop as it might be easier) optimizer? With maybe minimum assumptions it could shade some lights on this difference of behaviour.

## Typos:

Section 5: 'emprical'

Appendix S3.1: 'btheta' missing '' in tex file probably

## Overall

The author present a very interesting experimental approach to many Lipschitz claims about neural network and are convincing at exploring, discussing and interpreting their finding. As such I think it is a good paper. My main limitation for a conference is the fact that the paper is mostly located within its own appendix.

---

> ### Author Response · Authors · 2023-11-21
> **Response**
>
> Thank you for your positive review and thorough feedback. We are glad to hear that you found our paper covers the topic pretty well with an intense experimental protocol resulting in reasonable or theoretically supported interpretations. Below, we would like to take this opportunity to discuss in detail the concerns and questions raised by you:
>
> &nbsp;
>
> ## Limitation: Copiousness of the work
>
> That’s a quite understandable point. A work aiming to be comprehensive runs the risk of being obscure, and we tried our best to tread this tightrope as far as possible. But, much to our dismay, this seems to have come at the cost of causing the reader a back-and-forth between the main paper and the appendix.
>
> *Presentation strategies*. We will make our presentation more succinct and, more generally, try to reorganize some of the presentation which we were unable to do before. More concretely, we will focus on covering at least one scenario in each of the subsections in its completeness, so that the back-and-forth is minimized and if there is still some, it becomes more accessory. We believe at the moment there is some redundancy in the presentation of the figures. For instance, some of them can be combined or superimposed into one (surely, Figures 2 and 3; but potentially also the subfigures in Figures 1, 5, 7). This additional space will readily allow us to cut down the back-and-forth and make the overall read more comfortable and unhindered.
>
> *Journal Version*. This is a welcome suggestion. At some point down the line, we definitely plan to indeed further develop and exposit our paper through the journal format. But more than the cosmetic changes, we would like the journal version to also rigorously characterise the theoretical aspect of these findings and look, in more detail, into some other application areas (such as NLP). However, given that this would require a rather non-trivial amount of time and effort, we believe that it would be highly worthwhile to already disseminate the current set of mature (albeit empirical) findings — about an important topic like Lipschitz continuity —  in the scientific community via publishing at a highly relevant conference like ICLR.
>
> ## Other concerns.
>
> > Lack of theoretical insights … easy weakness for an experimental paper.
>
> - As you have justly recognized, our focus has primarily been on uncovering intriguing **empirical aspects** of the Lipschitz constant — which makes for an easy weakness in any experimental paper.
> - Also, we want to emphasize that such a focus is a natural consequence of our approaching this lofty problem while being grounded in empirical observations since the other direction of landing groundward from theory can often either overshoot or even be amiss.
> - Nevertheless recognizing the value of theoretical insights, we have attempted to offer, wherever possible, useful *theoretical arguments to complement these insights*, but surely, they are *not yet definitive*.
>
> Thus, given the current scope, we were forced to leave a more in-depth theoretical investigation of the presented phenomena as well as the matured empirical insights to a future version of our work.
>
> *Bibliographic Issues*: We have fixed these issues. The issue about (Gomez et al. 2020) not being (Latorre et al.)  stems from the full name of its first author, i.e., Fabian Latorre Gómez, from the DBLP profile https://dblp.uni-trier.de/pid/244/9638.html.
>
> *Typos*: These have been fixed too. Thanks for pointing out.

---

> ### Author Response · Authors · 2023-11-21
> **Response (continued)**
>
> ## Regarding the questions
>
> 1. We have changed Definition 2.2 to a Proposition, it does indeed make more sense to name it so.
>
> 2. About convex combinations and relation to points sampled via a Gaussian:
>
> *Geometry of convex combinations*. We compute convex combinations as a weighted (Euclidean) mean between two input points in the dataset. There can be two scenarios roughly: (a) When the considered points are distant, the convex combination (based on Euclidean mean) may not completely lie on the data manifold. (b) However, for closeby points, the convex combination should still yield a data point that is close to the data manifold, if not on it.
>
> So, it is a bit different from looking at random points around samples, which will yield points closer to the data manifold. But, in practice, as we consider a very large number of convex combinations, we expect to cover both these scenarios and the convex combinations should obtain results at least as good as that obtained through random sampling,  in terms of the lower Lipschitz bound.
>
> *Intuition*: As shown elsewhere, we can expect the highest Jacobian norms on points near the decision boundary, i.e., difficult points on the data manifold. Hence, in general, we think that your intuition is reasonable.
>
> *Empirical observations and hypothesis*. However, our empirical data suggest that sampling convex combinations results in larger $C_{lower}$ estimates. We refer to Figures 1 and S5 for this: even though S5 shows a Double Descent scenario, it shows that for most models the lower Lipschitz estimate barely changes when samples with Gaussian noise are considered. We guess this happens since we sample points from an isotropic Gaussian, of varying standard deviation, which consequently does not properly factor in the data covariance. In contrast, convex combinations would yield a point that better preserves the structure of the data covariance.
>
> 3. In Figure 8, the practical variance generally follows the theory. They are not fully identical, but the curve with $C_{lower}$ is rather close.
>
> 4. In case you might not have noticed, we do already show some results with a Vision Transformer (ViT) in Figures S7 and S16.  But, nevertheless, studying this in the context of a language modelling task would further the relevance of our work, and we intend to provide such experiments in a future expanded (journal) version of the paper. For now, given the architectural similarities, Figures S7 and S16 might point towards the nature of the behaviour that we can expect to see.
>
> &nbsp;
>
> ## Regarding the Appendix
>
> 1. > S4, Intermediate-points based analysis: … the supremum should be considered on all $\theta$
>
> That’s a good observation, and ideally, we could do that. We didn’t opt for it because:
>
> (a) As the theoretical analysis considers all encountered iterates in the optimization trajectory, so the distance between consecutive iterates is usually small enough such that quadratic or higher order terms in this distance are negligible and the first-order Taylor approximation $|f(\theta_{t+1}, x) - f(\theta_t, x)| \approx \nabla_\theta f(\theta_t, x)^\top (\theta_{t+1} - \theta_t)$ holds. This together with Cauchy-Schwarz should explain the choice of $C^{\text{discrete}}$. We will make this more explicit.
> (b) Practically, this step also allows  **make our estimates tractable**. We aimed to work out a bound that could potentially be estimated with our Lipschitz bounds and evaluated on real data (which we did in Appendix S3.4).
>
> To sum up, as you have pointed out, this isn’t critical. Even if we considered the whole set of $\theta$ values, then we would arrive at a result similar about the growth of the Lipschitz during training.
>
> 2.
> > S 3.17 such a marginal difference? [Lipschitz with MSE marginally lower than with CE]
>
> We attribute this difference to the effects of Softmax and the nature of CE loss. When minimizing the network parameters via CE, the solution is found at $\infty$, i.e., by driving the parameter norms $\rightarrow \infty$ so as to crank up the logits resulting in a one-hot softmax output. In practice, when training for a large but finite number of epochs, the logits end up being quite large. So then, Figures 28a and 28b can be understood as:
> - Figure 28a: When the function definition (and thus the lower Lipschitz) does not include the Softmax operation, the Lipschitz values are large since the logits (and parameter norms) are large.
> - Figure 28b: If the function definition includes Softmax, the rescaling of the outputs achieved via Softmax, suppresses the Lipschitz value as compared to Figure 28a (without Softmax), and brings it closer to the eventual value achieved by MSE.
> - MSE imposes a stronger, more direct, constraint (forcing predictions, here for classification tasks, to be exactly one-hot), so it results in still marginally smaller Lipschitz values relative to above than the ‘softer’ constraint via CE loss.

---

> ### Author Response · Authors · 2023-11-21
> **Response (continued, part 2)**
>
> 3. >  S3.18: Have you try to apply the same methods as in S4.1 but for Adam (or maybe RMSprop as it might be easier) optimizer?
>
> That is a very interesting suggestion, and which indeed could form for a quite fruitful pursuit! But, carrying it out, with the quality we would like to strive for, would be outside the current confines. This will, undoubtedly, be in the top few of our list of questions for further consideration.
>
> &nbsp;
>
> *We hope that our above response comprehensively addresses your remaining concerns. However, if something remains unclear or if there are any further questions, we are happy to answer them.*

---

> > ### Comment · Reviewer_F3fH · 2023-12-04
> > **Rebuttal**
> >
> > Thank you very much for the detailed response to all the points I raised, it greatly clarified all my questions. I will however maintain and confirm my grade of '8' on this paper.

---

### Author Response · Authors · 2023-11-21
**General response**

Dear reviewers and AC,


&nbsp;


We would like to thank all of you for evaluating our paper and providing thorough feedback on our work. We were glad to see that most reviewers rated the paper positively, in that the paper is **likely to attract substantial interest to the ML community** (reviewer Jyw9), contains **“extensive experiments”** (reviewer EeTb, Jyw9),  which are **“quite convincing”** (reviewer F3fH) and can offer an **“interesting perspective”** (reviewer Sk7T), and that the paper, in general, is **“very nice to read”** (reviewer F3fH).


## Individual responses and paper revision
Simultaneously,  we made our best efforts to address the leftover concerns, and have therefore written individual responses and have updated the paper. *All major additions are colored in **green** for easier navigation*.

## Fresh experiments
Firstly, we would like to thank the reviewers for their patience, as it took us some while to ensure we had all the requested experiments.
- **S3.21, S3.22, page 41-42**: In particular, we have also added requested experiments exploring the effect of weight and dropout on Lipschitz behaviour of neural networks.
- **S3.23, page 42**: Moreover, we also present the results of the lower Lipschitz bound separately over misclassifed and correctly classified experiments.




&nbsp;


We are confident that the reviewer feedback and the incorporated experiments has even further strengthened our paper, and that the reviewers will have a similar sentiment. Lastly, we are looking forward to hearing from you and we remain at your disposal should you have any comments/suggestions.

&nbsp;

Authors

---

> ### Author Response · Authors · 2023-11-22
>
> We thank all the reviewers for their engagement. A final concern that seems to be brewing is around the suitability of our paper to ICLR, given its lengthy appendix.
>
> &nbsp;
>
> To put it simply, *we believe that a thorough empirical exploration, complemented with theoretical arguments, on a core topic such as a Lipschitz continuity of deep neural networks has a strong thematic fit to ICLR. We will also be able to reach a wider audience and disseminate our findings to both the applied (adversarial samples, Lipschitz regularization) and theoretical (generalization) deep learning community adequately via a conference such as ICLR.*
>
> &nbsp;
>
> But, to answer this in more detail,
>
> - *The length of the appendix:* We admit that the paper ended up being a bit lengthy, but this was essentially due to providing **additional context for a broader readership**, including supplementary experiments, that are relevant to the topic and which a reader might feel curious about (like, for instance, the recent experiments asked by reviewer Jyw9 around weight decay and dropout), and providing sufficient experimental details to ensure reproducibility.
>
> &nbsp;
>
> - *The perception of the paper located within its appendix:*
>
>    - It is also true that we sometimes leave several details in the Appendix, but we believe most of these details are not as crucial and are secondary to the larger point that is being made in the corresponding section. However, please also see the **presentation strategies** we highlighted in this comment to Reviewer F3fH https://openreview.net/forum?id=5jWsW08zUh&noteId=g77tbAoKNJ, which will afford us more space in the main text and let us integrate these details within it, and thus *reduce the back-and-forth to the minimum*.
>
>    - Besides, one could also levy a similar argument, of the paper being located within the appendix, on theoretical works, which mainly comprise pages of long proofs (that is arguably the core contribution of such works) located usually within the appendix, which would however sound inappropriate.

---

### Meta-Review · Area_Chair_yULV · 2023-12-15

**Metareview:**

Lipschitz constants of deep networks show up in a variety of contexts and the work studies a few pertinent questions related to this. The reviewers mostly feel (and the AC agrees) that the questions are clearly articulated and arguably well justified. The focus of the study is empirical, and sufficient empirical evidence has been provided in support of the questions/hypotheses.

Such work can face standard critiques/concerns, e.g., why no theory, why not more experiments, etc., and the work has received such questions. The authors have responded in detail and added additional experiments. The reviewers have mostly engaged in the discussion.

In summary, the work does advance aspects of our understanding on an important theme in the context of deep learning.

**Justification For Why Not Higher Score:**

The support from some of the reviewers were a bit luke-warm as possibly more can be/needs to be done, there are parallel ongoing work (e.g., on double descent), etc.

**Justification For Why Not Lower Score:**

The work does advance aspects of our empirical understanding on an important theme in the context of deep learning.

---

### Decision · Program_Chairs · 2024-01-16

Accept (poster)